SciPost Physics

Submission

# Viscous dissipation of surface waves
# and its relevance to analogue gravity experiments

S. Robertson[1*], G. Rousseaux[2]

**1** Laboratoire de Physique Théorique, UMR 8627, CNRS,
Univ. Paris-Sud, Université Paris-Saclay, 91405 Orsay, France
**2** Institut Pprime, UPR 3346, CNRS-Université de Poitiers-ISAE ENSMA,
11 Boulevard Marie et Pierre Curie-Téléport 2, BP 30179, 86962 Futuroscope, France
* scott.robertson@th.u-psud.fr

March 15, 2018

## Abstract

**We consider dissipation of surface waves on fluids, with a view to its effects on analogue gravity experiments. We begin by reviewing some general properties of wave dissipation, before restricting our attention to surface waves and the dissipative role played by viscosity there. Finally, with particular focus on water, we consider several experimental setups inspired by analogue gravity: the analogue Hawking effect, the black hole laser, the analogue wormhole, and double bouncing at the wormhole entrance. Dissipative effects are considered in each, and we give estimates for their optimized experimental parameters.**

# 1  Introduction

Analogue gravity grew out of Unruh's simple but profound observation of 1981 [1]: the propagation of waves on a curved spacetime metric is (in the long-wavelength limit) mathematically analogous to wave propagation in a non-uniform medium, the "curvature" being supplied by the spatial and/or temporal variations of the background. In particular, analogue event horizons can exist in moving media, and are predicted to emit radiation via a process analogous to the black-hole evaporation discovered by Hawking [2]. Once it was understood that analogue Hawking radiation is largely insensitive to the short-wavelength dispersion [3–6] ubiquitous in realistic media, there was a surge of interest in the field. To date, there have been theoretical and experimental studies in many physical systems, including optics [7, 8], BEC [9, 10], polaritons [11–13], spin waves [14, 15], air [16, 17], and surface waves on water [18–26]. The dispersive corrections to the analogue Hawking flux have been well studied (see e.g. [27, 28]), providing theorists with a set of tools for treating short-scale Lorentz violation (due e.g. to the quantization of spacetime) and some insight into the possible effects thus obtained.

Another mechanism through which the purely relativistic scenario can be altered is dissipation [29–32]. This is just as ubiquitous as dispersion, affecting all of the analogue gravity systems mentioned above. In BEC, for example, phonons are subject to Landau-Beliaev damping due to interactions with the non-condensed part of the cloud [33]. Optical media are all absorptive to some extent, while for polaritons the main dissipative channel is the radiation of photons to the environment [11, 12]. Spin waves are subject to losses through ohmic dissipation [14, 15]. In fluids, the main source of dissipation is viscosity, though this in turn engenders several types of dissipative effects, such as absorption in the bulk and friction at the boundaries. Wave breaking can be considered as a 'super-dissipative' mechanism, with wave energy being lost to turbulence. Even astrophysical black holes may be subject to dissipative effects via quantum fluctuations of the space-time geometry [34, 35]. Dissipation must therefore be taken into account in the design of experiments if we are to maximize our chances of observing the effects of interest.

In this paper, we consider dissipation of surface waves on fluids, with the aim of determining its relevance to several experiments inspired by the analogue gravity program. Our focus shall be on minimizing the overall dissipative effects, rather than on a quantitative characterization of the corrections induced by dissipation (as done, e.g., in [32]). When discussing possible experiments, we give particular emphasis to surface waves on water, while in the more general discussions on viscous dissipation we write expressions in an adimensionalized form that can apply to many different fluids. Our hope is that the paper will pique the interest of experimentalists in fluid mechanics, who may be unfamiliar with the analogue gravity program.

The paper is organised as follows. In Section 2 we consider dissipation in a general sense, reviewing its effects on the dynamical evolution of a physical system. In Section 3 we focus

on the specific system of surface waves on fluids, reviewing the role of viscosity in providing a dissipative mechanism there. Finally, in Section 4 we turn to surface wave experiments in analogue gravity, optimizing their experimental design by attempting to minimize the identified dissipative effects. We summarize and conclude in Section 5.

# 2 General properties of dissipation

In this section, we review the properties of dissipative oscillatory systems through consideration of a few simple examples. Our goal is to pinpoint the generic features introduced by dissipation, in order to anticipate and understand the role it plays in the behavior of surface waves.

## 2.1 The damped pendulum

Since the stationary modes of a non-dissipative wave equation behave like harmonic oscillators, it is instructive to begin with a brief review of the one-dimensional damped harmonic oscillator. In adimensionalized units, its equation of motion is

$$\ddot{q} + 2\zeta\dot{q} + q = 0 \,, \tag{1}$$

where $\zeta \geq 0$ and where the overdot signifies a derivative with respect to time. This might, for example, represent the position $q$ of a mass on a spring, subject to a dynamic frictional force proportional to $-\zeta\dot{q}$. Although the coordinate $q$ is a real quantity, the fact that Eq. (1) is linear and invariant under complex conjugation means that $q$ can be expressed as the real part of a complex solution. Making the ansatz $q(t) = \mathrm{Re}\left\{\mathcal{A}e^{-i\omega t}\right\}$ where $\mathcal{A}$ is a complex number, and plugging this into Eq. (1), we find that $\omega$ must satisfy

$$\omega^2 + 2i\zeta\omega - 1 = 0 \qquad \Longleftrightarrow \qquad \omega = \pm\sqrt{1 - \zeta^2} - i\zeta \,. \tag{2}$$

The frequency is thus made complex by the presence of $\zeta$, which characterizes the strength of the damping. The negative imaginary part of $\omega$ causes the amplitude of the oscillations of $q$ to decrease exponentially with time.

It is instructive to consider the limits of small and large $\zeta$:

$$\omega \approx \pm 1 - i\zeta \qquad\qquad \text{for} \quad \zeta \ll 1 \,, \tag{3a}$$

$$\omega \approx -i\left(2\zeta\right) \quad \text{or} \quad -i/\left(2\zeta\right) \qquad \text{for} \quad \zeta \gg 1 \,. \tag{3b}$$

For very small $\zeta$, the imaginary part of $\omega$ is simply proportional to $\zeta$ while the real part is unaffected. This is the underdamped regime: the pendulum still oscillates at frequency 1, but the amplitude of the oscillations gradually decreases in time according to the exponential factor $e^{-\zeta t}$. As $\zeta$ increases, one notices a second-order shift in the real part of the frequency; indeed, the oscillation period is $2\pi/\sqrt{1 - \zeta^2} \approx 2\pi\left(1 + \zeta^2/2\right)$ for small $\zeta$. For $\zeta \to 1$, the period goes to infinity, and the two roots merge; this is the point of critical damping, corresponding to a saddle-node bifurcation in the behavior of the roots [36]. Increasing $\zeta$ even further, the roots are now both negative imaginary, and simply describe a non-oscillating pendulum returning to its equilibrium position at a well-defined rate. This is the overdamped regime, and unless fine tuning is applied, the slower of the two exponential rates will dominate the late-time behavior, so that the approach to equilibrium becomes slower with increasing $\zeta$.

## 2.2   Acoustic waves

Let us now consider a simple example of a dissipative wave equation. For simplicity (and with later application to a water channel in mind), we consider only one spatial dimension. Acoustic waves in a gas obey an equation of the form [37]

$$\partial_t^2 \phi - c^2 \partial_x^2 \phi - \frac{4}{3} \nu \partial_t \partial_x^2 \phi = 0 \,, \tag{4}$$

where $c$ is the wave speed in the limit $k \to 0$ and $\nu$ is the gas viscosity. As for the pendulum, the linearity of Eq. (4) and its invariance under complex conjugation allows us to treat $\phi$ as a complex variable. In particular, we can look for stationary modes of the form $e^{ikx - i\omega t}$. Plugging this ansatz into Eq. (4), we find that $\omega$ and $k$ must satisfy

$$\omega^2 - c^2 k^2 + i \frac{4}{3} \nu k^2 \omega = 0 \qquad \Longleftrightarrow \qquad \begin{cases} \omega = \pm \sqrt{c^2 k^2 - \left( \frac{2}{3} \nu k^2 \right)^2} - i \frac{2}{3} \nu k^2 \\[2mm] k = \pm \frac{\omega}{c} \left( 1 - i \, 4\nu\omega/3c^2 \right)^{-1/2} \end{cases} . \tag{5}$$

All three equations here are entirely equivalent. The relative appropriateness of the explicit expressions on the right-hand side depends on the form of the boundary conditions imposed on the system, as we now discuss.

Firstly, the equation giving $\omega$ as a function of $k$ is very much akin to the right-hand side of (2) for the frequency of the damped pendulum; in particular, it has a saddle-node bifurcation at $2\nu |k| / 3c = 1$, so that $2\nu |k| / 3c$ plays the role of the pendulum damping parameter $\zeta$. This similarity is not an accident: taking a plane wave in space so that $-\partial_x^2 \phi = k^2 \phi$, Eq. (4) has exactly the same form (up to the adimensionalization of the time coordinate $t$) as Eq. (1). The essential difference is the occurrence of the variable $k$, which can be thought of as labelling different modes in the spectrum that evolve independently of each other. This equation would be used to describe the solution if the gas were placed in an initial state $(\phi(x), \partial_t \phi(x))$ and allowed to evolve freely: the solution can be resolved (via Fourier analysis) into a series of plane waves of real $k$, each of which oscillates and/or dissipates in time according to the first equation on the right of (5).

The second equation on the right of (5), giving $k$ as a function of $\omega$, is appropriate for describing the response to a different type of boundary condition, one that forces the system to oscillate in time with a given frequency. Suppose that, through some periodic driving mechanism applied at a certain position, the system is put into a stationary state of frequency $\omega$; for surface waves, this is typically what is done in practice through the placement of a wave maker at one end of the flume. Note that here $\omega$ is real (there is no decrease of the amplitude with time), and so this situation is unlike the previous case in which the wave number $k$ is taken to be real and the damping rate is encoded by the (complex) solution for $\omega$. Instead, the dissipative effects will be seen as attenuation in space, as the amplitude of the resulting oscillations will decay as one moves further from the source. We are therefore looking for (complex) $k$ as a function of (real) $\omega$, and this is exactly the content of the second equation on the right of (5). Interestingly, in contrast to the case of real $k$, there is no singular point along the real $\omega$-axis, so the system evolves smoothly between the underdamped and overdamped regimes. Note also that, despite the presence of the $k^4$ term in the expression for $\omega(k)$, there are only two wave vector solutions for any given $\omega \neq 0$. This shows that, at

least when dissipation becomes strong, the real part of $\omega(k)$ is insufficient to determine the number of roots at any given frequency.

As for the pendulum, it is again of interest to consider the limits of weak and strong damping. For the first of Eqs. (5), the results are analogous to those of the pendulum:

$$\omega \approx \pm ck - i\frac{2}{3}\nu k^2 \qquad \text{for} \qquad \text{Re}_k \gg 1, \tag{6a}$$

$$\omega \approx -i\frac{4}{3}\nu k^2 \quad \text{or} \quad -i\frac{3c^2}{4\nu} \qquad \text{for} \qquad \text{Re}_k \ll 1, \tag{6b}$$

where $\text{Re}_k \equiv 3c/2\nu k$ is a Reynolds number formed from the wave number. We can look similarly at the limits for the second of Eqs. (5):

$$k \approx \pm \left[ \frac{\omega}{c} + i\frac{2}{3}\frac{\nu}{c}\left(\frac{\omega}{c}\right)^2 \right] \qquad \text{for} \qquad \text{Re}_\omega \gg 1, \tag{7a}$$

$$k \approx \pm (1 + i)\sqrt{\frac{3\omega}{8\nu}} \qquad \text{for} \qquad \text{Re}_\omega \ll 1, \tag{7b}$$

where $\text{Re}_\omega \equiv 3c^2/4\nu\omega$ is a Reynolds number formed from the frequency. For real frequencies, the wave number never becomes purely imaginary, but its real and imaginary parts do become equal in the limit of large damping, so that the wavelength becomes ill-defined. Equations (7a) and (7b) respectively describe the underdamped and overdamped regimes as seen from the real $\omega$-axis.

Whether we express $\omega$ as a function of $k$ or vice versa, it is always the case that, when the damping is weak (or equivalently, when the relevant Reynolds number is large), the dissipation manifests itself as a small imaginary correction to the frequency ($\omega \to \omega_r - i\Gamma$) or wave vector ($k \to k_r + i\Delta$), where $\omega_r$ and $k_r$ (as well as $\Gamma$ and $\Delta$) are defined to be real. These two corrections, $\Gamma$ and $\Delta$, are related in a physically intuitive way [1] via the group velocity $v_g = d\omega/dk$, which remains well-defined in the limit of weak damping. The dissipative rate being $\Gamma$ means that the amplitude of the wave is reduced after a time interval $\delta t$ by the factor $e^{-\Gamma \delta t}$. Knowing the group velocity $v_g$ of the wave (this being the propagation speed of the amplitude envelope whenever it varies with $x$), the time interval $\delta t$ is related to the propagation distance $\delta x$ by $\delta t = \delta x/v_g$, and therefore, the attenuation of the wave in space follows the exponential factor $e^{-\Gamma \delta x/v_g}$. We thus have $\Delta = \Gamma/v_g$, and a close comparison of Eqs. (6a) and (7a) shows that they are related in precisely this way. Note that, while $\Gamma$ is necessarily positive (if the wave amplitude is to decrease with time), the sign of $\Delta$ depends on the direction of the group velocity: its sign is always the same as the sign of $v_g$, so that the wave amplitude decreases in the direction of propagation.

Finally, having defined the spatial attenuation coefficient $\Delta$, it is also convenient to define the dissipative length $l_\text{diss}$ such that the spatial attenuation of the wave amplitude [2] follows (in

---

[1] A more precise procedure is to consider a stationary situation in which $\omega$ is exactly real, while $k = k_r + ik_i$ has a non-zero imaginary part. The plane wave solution is thus $e^{ikx} = e^{ik_r x - k_i x} \equiv e^{ik_r x} e^{-\Delta x}$, so by definition $\Delta = k_i$. We can approximate $k_i$ by writing $\omega - i\Gamma = \omega(k_r) = \omega(k_r + ik_i - ik_i) \approx \omega(k_r + ik_i) - ik_i \omega'(k_r + ik_i)$. Since $\omega(k_r + ik_i)$ is real by definition, we find that $\Gamma = k_i \omega'(k_r + ik_i) = k_i v_g(k_r) + O(k_i^2)$, and therefore $\Delta \approx \Gamma/v_g(k_r)$. This procedure indicates that the expression for $\Delta$ is valid so long as $|k_i/k_r| \ll 1$, since it is only then that the first-order Taylor expansion of $\omega(k_r)$ performed above is valid.

[2] Since the wave energy varies as the square of the amplitude, the energy is attenuated according to $e^{-2|\Delta x|/l_\text{diss}}$. We could thus define a dissipative length associated to the energy: $l_\text{diss}^E \equiv l_\text{diss}/2$, so that $E \propto e^{-|\Delta x|/l_\text{diss}^E}$. In this paper, we only consider $l_\text{diss}$ associated to the attenuation of the wave amplitude.

the propagation direction) $e^{-|\Delta x|/l_{\mathrm{diss}}}$, i.e. $l_{\mathrm{diss}} = 1/\left|\Delta\right| = \left|v_g\right|/\Gamma$ and is taken to be positive. The dissipative length thus describes how far the wave travels before being significantly dissipated, and there are two ways in which it can become small: either $\Gamma$ becomes large so that the wave is dissipated very quickly; or $v_g$ becomes small so that, even though the dissipative rate may not be very large, the wave propagates so slowly that it is significantly damped even over a short distance.

## 2.3   The rotating conducting cylinder

The behavior of acoustic waves in a viscous gas encapsulates many of the qualitative features we shall encounter when considering surface waves on viscous fluids. For completeness, however, and to illustrate the diversity of effects that can be induced by dissipation, let us briefly mention the case of ohmic dissipation in a rotating cylinder, as originally studied by Zel'Dovich [38]. (For similar effects in acoustics, see Ref. [39].) The wave equation satisfied by the component of the vector potential parallel to the axis of the cylinder (which we here label the $z$-axis) is

$$(\partial_t + \Omega_{\mathrm{rot}}\partial_\phi)^2 A_z - c^2 \nabla^2 A_z + \mu_0 \sigma \partial_t A_z = 0\,, \tag{8}$$

where $\Omega_{\mathrm{rot}}$ is the angular velocity of the cylinder and $\sigma$ is its conductivity. As before, we can use a complex solution for $A_z$, which we take to be of the form $e^{-i\omega t + im\phi}f(r,z)$ where $\nabla^2\left(e^{im\phi}f(r,z)\right) = -k^2 e^{im\phi}f(r,z)$. Solving for $\omega$, we have

$$(\omega - \Omega_{\mathrm{rot}}m)^2 - c^2 k^2 + i\mu_0\sigma\omega = 0$$

$$\Longleftrightarrow \qquad \omega = \pm\sqrt{c^2 k^2 - \left(\frac{\mu_0\sigma}{2}\right)^2 - i\mu_0\sigma\Omega_{\mathrm{rot}}m} + \Omega_{\mathrm{rot}}m - i\frac{\mu_0\sigma}{2}\,. \tag{9}$$

Apart from the fact that this dispersion relation shows overdamping at low wave numbers rather than at high wave numbers as in Eqs. (5), the most interesting aspect of Eq. (9) is that, for $\Omega_{\mathrm{rot}}m \neq 0$, one of the solutions in the low-$k$ regime has a *positive* imaginary part. [3] This corresponds to a solution which grows in time, and hence to an instability.

That the inclusion of an ostensibly dissipative term can lead to instabilities was examined by Heisenberg in the context of fluid flows [40]. It is related to the process of superradiance [25, 26, 41–45], and is made possible by the simultaneous existence of damping and negative-energy waves [46, 47], the former occurring either in the bulk (as in the conducting cylinder) or across an event horizon (as in a rotating black hole), and the latter being induced by the motion of the system. Although this is not one of the setups we shall examine in Sec. 4, the notion of negative-energy waves and the instabilities they can induce is of direct relevance to the analogue Hawking effect and the black hole laser.

## 3   Viscous dissipation of surface waves

In this section we turn our attention to surface waves on fluids. We review their dispersion relation in the absence of dissipation, then examine the changes brought about by viscosity

---

[3]To see this, let $k = 0$ and write $\sqrt{-\left(\mu_0\sigma/2\right)^2 - i\mu_0\sigma\Omega m} = i\mu_0\sigma/2\sqrt{1 + i\,4\Omega m/\mu_0\sigma}$. It is straightforward to see geometrically that the real part of $\sqrt{1 + i\,4\Omega m/\mu_0\sigma}$ must be larger than 1, and therefore, one of the two solutions on the right of (9) has a positive imaginary part. By continuity, there must be a window around $k = 0$ where the imaginary part of the frequency is positive.

due both to friction at the boundaries of the channel and within the bulk of the fluid itself. We end the section by introducing a dissipative wave equation that includes the main effects of viscosity in the bulk, which is later used in numerical simulations of analogue gravity experiments in Section 4.

## 3.1   Dispersion relation on inviscid fluids

For linear surface waves on a static, inviscid fluid, the dispersion relation $\sigma(k)$ is given by [48, 49]

$$\sigma^2 = \left(gk + \frac{\gamma}{\rho}k^3\right)\tanh\left(hk\right), \tag{10}$$

where $g$ is the acceleration due to gravity, $\gamma$ is the surface tension of the fluid, $\rho$ is its density, and $h$ is its depth. The dispersion relation (10) divides naturally into the gravity and capillary branches, respectively defined according to whether $gk$ or $\gamma k^3/\rho$ is the larger term, and hence according to whether gravity or surface tension provides the dominant restoring force. Following LeBlond and Mainardi [50], it is convenient to define $k_\gamma$, $\sigma_\gamma$ and $u_\gamma$ as the wave number, frequency and phase velocity of the wave at which these two terms are exactly equal:

$$k_\gamma = \sqrt{\frac{g\rho}{\gamma}}, \qquad \sigma_\gamma = \sqrt{2gk_\gamma}, \qquad u_\gamma = \frac{\sigma_\gamma}{k_\gamma}. \tag{11}$$

The corresponding wavelength $2\pi/k_\gamma$ is referred to as the *capillary length* (in water it is about 1.7 cm, while its period $2\pi/\sigma_\gamma$ is about 74 ms). Using $k_\gamma$ and $\sigma_\gamma$, the dispersion relation (10) can be adimensionalized: we define $K = k/k_\gamma$, $\Sigma = \sigma/\sigma_\gamma$ and $H = hk_\gamma$, in which case Eq. (10) becomes equivalent to

$$\Sigma^2 = \frac{1}{2}\left(K + K^3\right)\tanh\left(HK\right). \tag{12}$$

In the deep limit $HK \gg 1$, this becomes

$$\Sigma^2 \approx \frac{1}{2}\left(K + K^3\right). \tag{13}$$

The dispersion relation (13) is shown in the left panel of Figure 1, along with an example of the correction due to finite depth (12). In the right panel are shown the corresponding phase and group velocities, in units of $u_\gamma$. We note in particular that $u_\gamma$ is the minimum phase velocity [4], and that the minimum group velocity is slightly less than this at $u_c \approx 0.768\,u_\gamma$ (in water these are equal to 23.1 cm/s and 17.8 cm/s, respectively).

---

[4]It is possible to get a smaller phase velocity at $K \to 0$ if $H$ is very small. In fact, since the adimensionalized phase velocity at $K \to 0$ is $\sqrt{H/2}$, this requires $H < 2$ (in water this is equivalent to $h < 5.5$ mm).

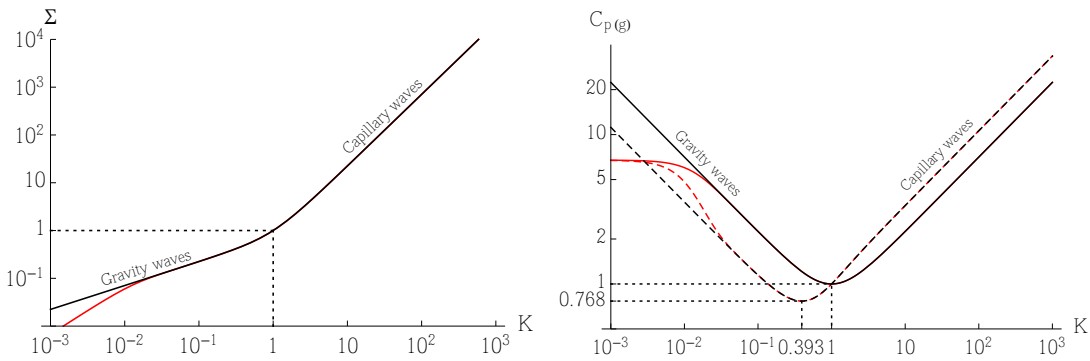

Figure 1: Surface wave dispersion on a static inviscid fluid. On the left is plotted, on a log-log scale, the adimensionalized dispersion relation $\Sigma(K)$; the black curve shows the dispersion relation in the limit of infinite depth (Eq. (13)), while the red curve shows the corrected dispersion relation when the depth is finite (Eq. 12)). Here, we have taken $H = k_\gamma h = 91$, which in water corresponds to a depth of $h = 25\,\text{cm}$. The gravity ($\Sigma \propto K^{1/2}$) and capillary ($\Sigma \propto K^{3/2}$) branches of the dispersion relation are clearly distinguished, as is the linear regime ($\Sigma \propto K$) when $HK \lesssim 1$. On the right are plotted the corresponding adimensionalized phase velocities $C_p$ (solid curves) and group velocities $C_g$ (dashed curves). Note that the minimum phase velocity is 1, occurring exactly at the crossover between the gravity and capillary branches (i.e. at $k = k_\gamma$, $\sigma = \sigma_\gamma$; see Eqs. (11)). The minimum group velocity is less than this: $U_c \approx 0.768$ at $K \approx 0.393$. When the dispersion relation becomes linear (as in the relativistic scenario) for $HK \lesssim 1$, the phase and group velocities approach the common value $\sqrt{H/2}$ (or $\sqrt{gh}$ in dimensionful units).

On a fluid which is not static but is flowing uniformly [5] at adimensionalized velocity $U \equiv u/u_\gamma$, we are free to perform a Galilean transformation to move into the "co-moving" frame in which the fluid is at rest, where dispersion relation (12) will hold. The frequency in the original "lab" frame is related to that in the co-moving frame by a Doppler shift, so that the dispersion relation in the former is

$$(\Sigma - UK)^2 = \frac{1}{2}\left(K + K^3\right)\tanh\left(HK\right) . \tag{14}$$

Examples of $\Sigma(K)$ for various values of $U$ are shown in the left panel of Figure 2. The phase and group velocities in the new frame are simply shifted by $U$. Note that, if $|U| > U_c \approx 0.768$ (the minimum group velocity on a static fluid, see Fig. 1), it will be strong enough to reverse the group velocities of some waves, leading to the existence of local extrema in the dispersion relation (since $v_g = d\omega/dk$) and hence to the existence of multiple roots for certain frequencies. Furthermore, when $|U| > 1$, even the phase velocity can be reversed, and since $v_p = \omega/k$ this means that the frequency itself has changed sign. Since the wave energy is proportional to $\omega$ [53], waves for which this happens have negative energy, i.e. their presence tends to reduce the total energy of the system. The velocity $u_\gamma$, then, is the surface wave analogue of the Landau critical velocity in superfluid physics [54], above which negative-energy excitations exist.

---

[5]We exclude in this analysis the possibility of flows which are non-uniform in the vertical and/or transverse direction, although it should be kept in mind that this is not entirely self-consistent since friction at the boundaries will tend to induce such non-uniformity; see e.g. Refs. [51, 52] for a treatment of flows which are non-uniform in the vertical direction.

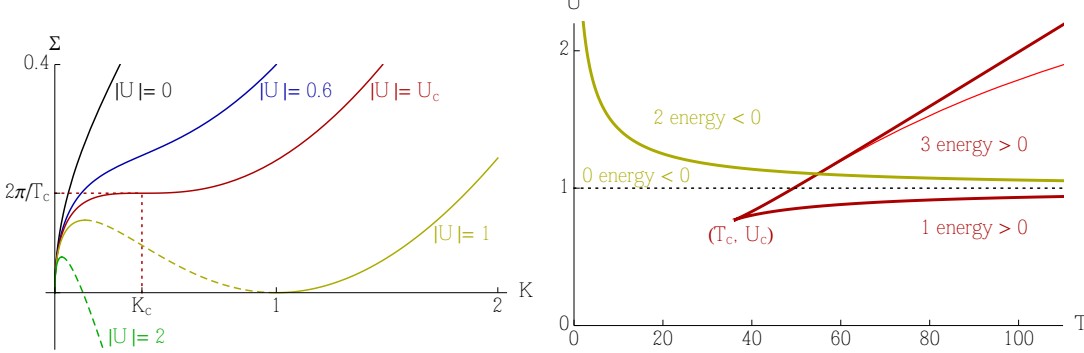

Figure 2: Dispersion relation of counter-propagating waves. In the left panel is plotted (in the deep limit $\tanh(HK) \to 1$) the adimensionalized frequency $\Sigma(K)$ for various flow velocities. The red curve corresponds to the critical value of $U$ at which the group velocity first vanishes for some value of $K$, as indicated by the point of inflection at $K = K_c \approx 0.393$. For larger $|U|$, the dispersion relation ceases to be one-to-one and the group velocity is negative in certain regions, which are indicated by the dashed curves. Moreover, when $|U| > 1$, a portion of the curve dips below 0 so that negative-frequency (and therefore negative-energy) solutions exist. In the right panel is shown the corresponding phase diagram [20] in the $(T, U)$-plane, where $T \equiv 2\pi/\Sigma$ is the adimensionalized period. The region within the red triangle is where the positive-energy branch of the dispersion relation becomes triple-valued, and a light red curve has been included to show how the diagram varies in the case of finite depth (here, we have taken $H = 32$, or $h = 8.7$cm in water). The yellow curve marks the boundary between the non-existence of negative-energy solutions (below the curve) and the existence of two of them (above the curve).

In the right panel of Fig. 2 is shown a phase diagram in the $(T, U)$-plane, where $T = 2\pi/\Sigma$ is the adimensionalized period (see Ref. [20] for further details). This plane is split into several regions, according to the number and type of wave vector solutions that exist. The red triangle is related to the number of positive-energy solutions: there is only one outside the triangle, while there are three inside. The yellow curve marks the boundary for the existence of negative-energy solutions: below it, there are no such solutions, while above it there are two. Note that this asymptotes to $U = 1$ when $T \to \infty$, in agreement with our claim above that this is the critical velocity above which negative-energy waves exist. The 'critical point' at the corner of the triangle occurs at precisely the flow velocity $U_c = 0.768$ above which the group velocity is reversed for some waves and the dispersion relation is no longer one-to-one.

## 3.2  Viscous boundary effects

Viscosity provides the principal mechanism by which wave energy is dissipated in fluids. It is essentially a friction force, i.e. a resistance to relative motion, and it manifests itself in two particularly important ways. One is a bulk effect, to which we turn in the next subsection. Here, we briefly focus on energy loss due to friction at the edges of the flume, a process of particular importance for long wavelengths which are sensitive to the boundaries of the system. For weak damping, the total damping rate will just be the sum of the two, i.e. $\Gamma_{\text{total}} = \Gamma_{\text{bulk}} + \Gamma_{\text{edges}}$.

Precise treatments give values of the spatial attenuation rate $\Delta_{\text{edges}}$ (and the dissipative

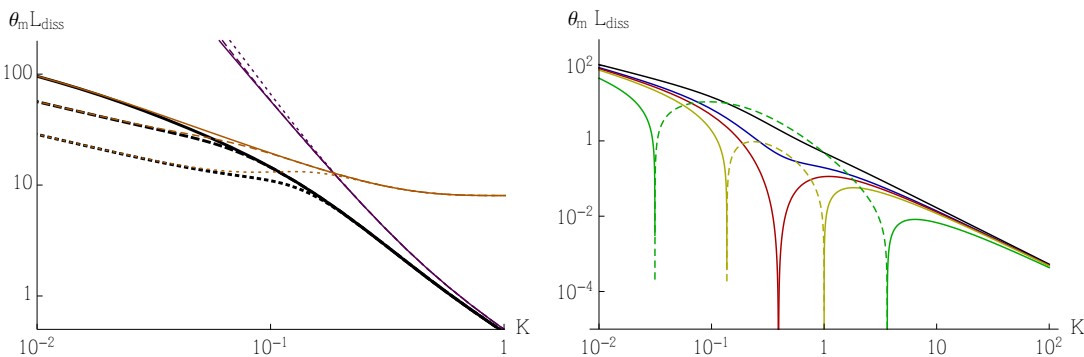

Figure 3: Adimensionalized dissipative length, as a function of wave vector $K$, associated to viscous damping. On the left is shown the dissipative length on a static fluid due to friction at the boundaries, with a channel width of $39\,\mathrm{cm}$ and water depths of $20\,\mathrm{cm}$ (solid curves), $10\,\mathrm{cm}$ (dashed curves) and $5\,\mathrm{cm}$ (dotted curves). The black curves show the combined effects of boundary friction and bulk viscosity; the orange curves are due to boundary friction alone, while the purple curves show the effect of bulk viscosity alone. On the right, the depth is fixed at $20\,\mathrm{cm}$ and both bulk viscosity and boundary friction are included (assuming for simplicity that Eqs. (15) remain valid in the rest frame of a uniform flow). The variously colored curves here correspond to the same flow velocities as in the left panel of Fig. 2. The dips where $L_{\mathrm{diss}} \to 0$ are points where the group velocity $d\Sigma/dK$ vanishes, and the dashed portions of the curves correspond to regions where $v_g$ is negative.

rate $\Gamma_{\mathrm{edges}}$) associated with loss at the boundaries, see [56–58]. In the gravity-wave regime and on a fluid which is itself at rest with respect to the physical boundaries, this is

$$\Delta_{\mathrm{edges}} = \sqrt{\frac{\nu}{2\sigma}} \frac{2k}{b} \frac{kb + \sinh(2kh)}{2kh + \sinh(2kh)} \qquad \Longleftrightarrow \qquad \Gamma_{\mathrm{edges}} = \sqrt{\frac{\nu\sigma}{2}} \left( \frac{1}{2h} \frac{2kh}{\sinh(2kh)} + \frac{1}{b} \right),$$
(15)

where $h$ is the depth and $b$ is the transverse width of the channel. Note the decomposition of $\Gamma_{\mathrm{edges}}$ into two distinct terms, the first of which vanishes as the depth $h \to \infty$ and the second of which vanishes as the width $b \to \infty$; these respectively represent the damping due to friction at the bottom of the flume and at the sides of the flume, i.e. $\Gamma_{\mathrm{edges}} = \Gamma_{\mathrm{bottom}} + \Gamma_{\mathrm{sides}}$. Notice that $\Gamma_{\mathrm{bottom}}$ tends rapidly to zero as $kh \to \infty$ while $\Gamma_{\mathrm{sides}}$ vanishes only as $1/b$. This is physically intuitive: surface waves are naturally evanescent in the vertical direction and thus quickly become insensitive to the fluid depth, whereas there is always a finite wave amplitude at the sides of the flume. Examples for a typical value of $b$ and various values of $h$ are shown in the left panel of Fig. 3.

Further corrections which can be made, and which are not considered in this paper, include the effects of non-uniformity of the flow due to boundary friction (see Refs. [51, 52] for a treatment of flows with a linear shear profile), and impurities on the surface which give rise to an effective boundary friction there [58, 59].

## 3.3   Dispersion relation in presence of bulk viscosity

Let us now turn to viscous dissipation occuring in the bulk of the fluid, i.e. to the friction generated by the relative movement of different fluid layers. For a static fluid of infinite

depth (i.e. $\tanh(hk) \to 1$), the exact expression for the dispersion relation of surface waves is known [48, 50]. [6] It is most conveniently expressed using the following adimensionalized quantities:

$$\theta_\gamma = \frac{\nu k_\gamma^2}{\sigma_\gamma}, \qquad \theta = \theta_\gamma \frac{K^2}{\Sigma(K)}, \qquad x = -i\frac{\Omega + 2i\theta_\gamma K^2}{\Sigma(K)}, \tag{16}$$

where $\Sigma(K)$ is the inviscid dispersion relation (13) and $\Omega = \Omega(K)$ is the (complex) dispersion relation in the viscous case, which is to be solved for. The parameter $\theta_\gamma$ is an adimensionalized form of the kinematic viscosity $\nu$, while $\theta = \theta(K)$ is a monotonic function of $K$ such that $\theta(K = 1) = \theta_\gamma$. [7] Solving for the dispersion relation $\Omega(K)$ is equivalent to solving for $x(\theta)$, which turns out to be the solution of the polynomial [48, 50]

$$\left(x^2 + 1\right)^2 = 16\,\theta^3\,(x - \theta), \tag{17}$$

with the additional constraint that $\mathrm{Re}\left\{x^2 + 1\right\} \geq 0$. In dimensionful quantities, this is equivalent to the implicit relation [49]

$$\left(\omega + 2i\nu k^2\right)^2 = \sigma^2(k) - 4\nu^{3/2}k^3\sqrt{\nu k^2 - i\omega}, \tag{18}$$

where $\mathrm{Re}\left\{\sqrt{\nu k^2 - i\omega}\right\} \geq 0$.

For our purposes, we shall mainly be interested in the limit of weak dissipation. When $\theta$ is very small ($\theta \lesssim 0.04$ or, in water, $\lambda \gtrsim 5 \times 10^{-3}$cm), the right-hand side of Eq. (17) can be neglected, and we are left with the solutions $x \approx \pm i$; or, using the last of Eqs. (16),

$$\Omega \approx \pm\Sigma - 2i\theta_\gamma K^2 \qquad \Longleftrightarrow \qquad \omega \approx \pm\sigma - 2i\nu k^2, \tag{19}$$

where in the right-hand equation we have replaced adimensionalized variables with their dimensionful counterparts. To this level of approximation, then, the dissipative rate $\Gamma$ takes the simple form of $2\nu k^2$ [48], while the real part of the frequency is unaffected by viscosity [8]. In the presence of a uniform flow, the real part of the frequency will be Doppler shifted as in Eq. (14), but since the wave number $k$ is the same in all Galilean frames, the dissipative rate $\Gamma$ is the same as on a static fluid.

The dissipative length associated with viscosity in the bulk is

$$l_{\mathrm{diss}} = \frac{|v_g|}{\Gamma} = \frac{|c_g + u|}{2\nu k^2}. \tag{20}$$

In the second equality, $\Gamma$ has been replaced by $2\nu k^2$ and the total group velocity $v_g$ has been decomposed into $c_g$, the group velocity of the wave in the rest frame of the fluid, and $u$, the velocity of the flow. As for other quantities, it is convenient to adimensionalize Eq. (20) so as to be applicable to many fluids. This is easily done by defining $L_{\mathrm{diss}} = k_\gamma l_{\mathrm{diss}}$ and replacing all quantities by their adimensionalized counterparts:

$$\theta_\gamma L_{\mathrm{diss}} = \frac{|C_g + U|}{2K^2}. \tag{21}$$

---

[6]See also [60] for nonlinear corrections, where the wave amplitude cannot be treated as infinitesimal.

[7]Note that $1/\theta_\gamma$ can be written as $u_\gamma/(\nu k_\gamma)$ which, up to a factor of $2\pi$, is $u_\gamma \lambda_m/\nu$. It thus takes the form of a Reynolds number, with the characteristic velocity and length scale taken as the phase velocity and wavelength of the gravity-capillary crossover. Similarly, $1/\theta(K)$ is a wave number-dependent Reynolds number.

[8]As already indicated in our brief treatment of acoustic waves, this is quite a general behavior; see also Eq. (36) of [61], where a similar expression describes the dissipation of inertial waves.

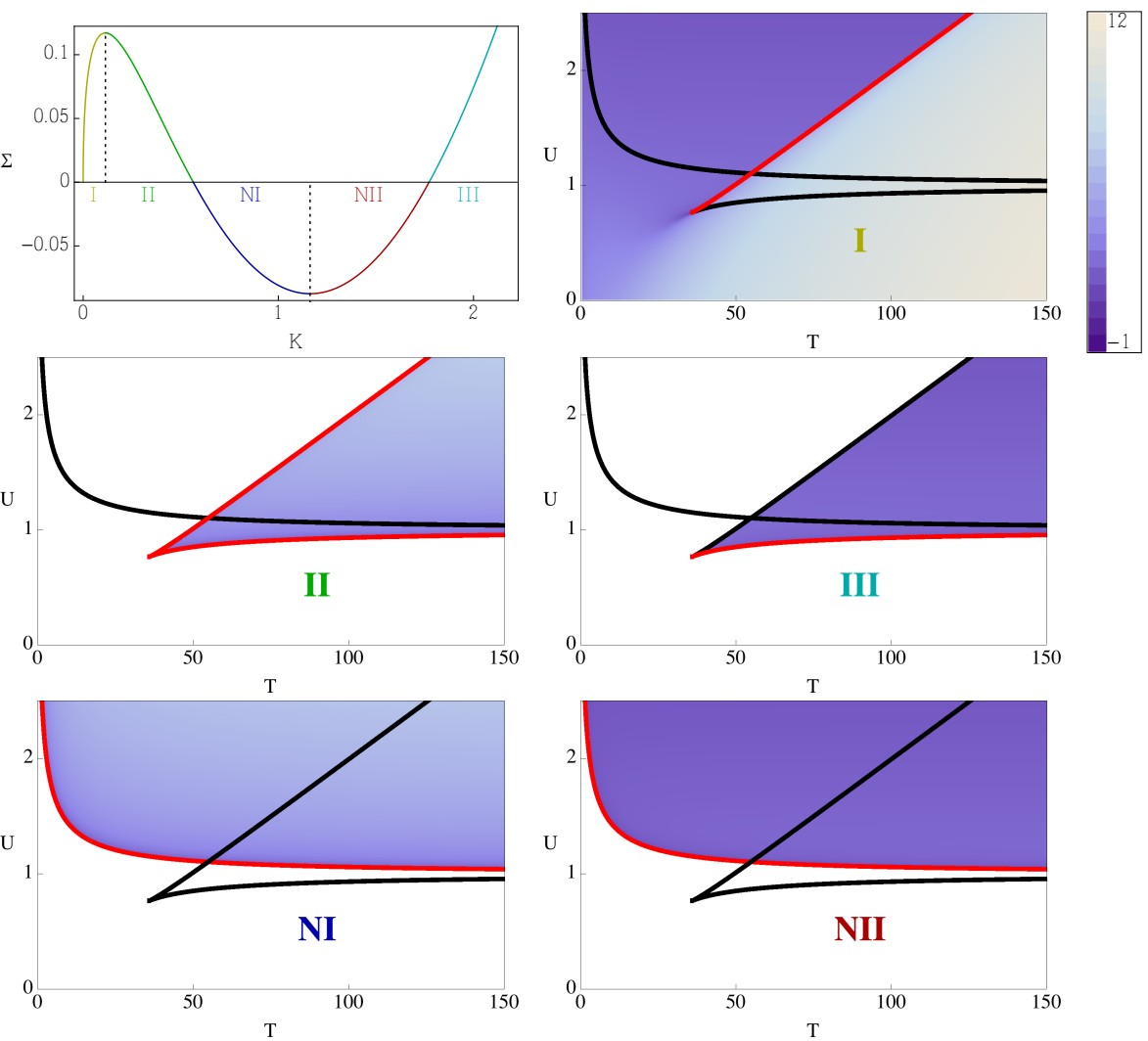

Figure 4: Dissipative lengths associated with bulk viscosity. The upper left plot indicates (at an adimensionalized flow velocity $U = -1.08$) the five different branches of the dispersion relation, with the positive-frequency solutions labelled *I-III* and the negative-frequency solutions labelled *NI-NII*. Roots *I-III* are arranged in order of increasing magnitude, with *I* and *III* having positive group velocity and *II* having negative group velocity. Similarly, roots *NI* and *NII* are also arranged in order of increasing magnitude, whenever the negative-frequency solutions exist. The remaining five plots indicate, on the same $(U, T)$ phase space from the right panel of Fig. 2, the behavior of $\ln(l_{\mathrm{diss}}/\lambda)$ for each of these roots, the logarithm being used due to the large variation in the order of magnitude of $l_{\mathrm{diss}}/\lambda$. The red curves show the cuts at which the solution is discontinuous, while the white regions show where the root in question does not exist at all. Note that root *I* is transformed from a long-wavelength to a short-wavelength mode upon analytic continuation around the critical point; as remarked in [20], this is reminiscent of the phase transition between liquid and gas. It can even be analytically continued further, crossing the red curve (in the top right panel) from above to reach root *III*.

Note that, having multiplied through by $\theta_\gamma$, we obtain an expression on the right-hand side of Eq. (21) that depends only on $K$ and $U$. Examples are shown in Figure 3, with and without boundary friction in the left panel, and with and without current in the right panel. In Figure 4 is shown, on the $(T, U)$ phase space and for each of the roots of the dispersion relation, the behavior of (the logarithm of) the dimensionless number $l_{\text{diss}}/\lambda = k l_{\text{diss}}/2\pi$, which measures how many wavelengths the wave can propagate before being dissipated by bulk viscosity.

## 3.4 Dispersive effects of viscosity

Let us return briefly to the full dissipative dispersion relation of Eqs. (17) and (18). We have already seen from rather simple examples (see Sec. 2) that, as well as introducing an imaginary correction to the frequency, dissipative effects like viscosity have at the same time an effect on the real part of the frequency, thus altering the dispersion profile. This is also the case for the viscous dissipation of surface waves. In particular, there is a bifurcation in the solutions of Eq. (17) at $\theta = \theta_{\text{crit}} \approx 1.31$, with waves being overdamped if $\theta > \theta_{\text{crit}}$. Since $\theta(k)$ is a monotonically increasing function of $k$, this means that there are no propagating wave solutions if the wavelength is too small; the energy of such waves will simply dissipate, much as the overdamped pendulum (see Sec. 2.1) simply returns to equilibrium without oscillating.

In effect, the adimensionalized viscosity $\theta_\gamma$ determines where this short-scale cut-off occurs relative to the capillary length, with the cut-off occurring at wavelengths much smaller than the capillary length when $\theta_\gamma$ is small, and at wavelengths much larger than the capillary length when $\theta_\gamma$ is large. Several examples are shown in Figure 5, with the real part of the adimensionalized frequency shown in solid curves [9] and the imaginary part in dashed curves. In water, for example, we have $\theta_\gamma = 1.58 \times 10^{-3}$ and $\theta = \theta_{\text{crit}}$ occurring at a wavelength of about $50\,\text{nm}$, much shorter than the capillary length of about $3\,\text{mm}$. In glycerine, on the other hand, $\theta_\gamma = 3.9$, and the short-scale cut-off occurs at a wavelength of about $3.5\,\text{cm}$, which is an order of magnitude larger than the capillary length at around $2\,\text{mm}$. So, in the case of large $\theta_\gamma$, viscosity kills off wave propagation while we are still in the gravity wave regime; the capillary wave regime can never be reached. [10]

---

[9]It is tempting to conclude from the sharp drop in the real part of the frequency near the cut-off that there is an additional wave vector solution for a given frequency. This is in fact not the case, as the imaginary parts of the low- and high-$k$ solutions are very different. This can be understood by recalling the two forms of the dispersion relation for acoustic waves on the right-hand side of Eqs. (5): as mentioned there, there are always only two values of $k$ corresponding to a given $\omega$, even though the real part of $\omega$ taken on its own would give four roots.

[10]This could have quite drastic consequences for the circular jump experiment of [55]. There, zero-frequency capillary modes typically appear, but they may be heavily suppressed if they occur close to or even beyond the dissipative dispersive cut-off.

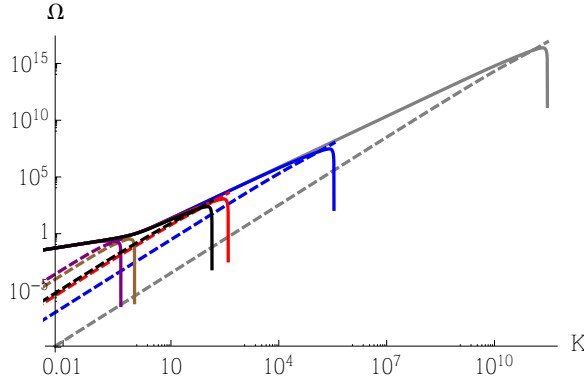

Figure 5: Viscous dispersion relation for surface waves on static fluids of infinite depth. The solid curves plot the real part of the frequency while the dashed lines show the imaginary part (but only for $\theta < \theta_{\mathrm{crit}}$). The different colors correspond to different fluids: mercury (grey, $\theta_\gamma = 1.7 \times 10^{-6}$), water (blue, $\theta_\gamma = 1.58 \times 10^{-3}$), glycol (red, $\theta_\gamma = 4.7 \times 10^{-2}$), silicone oil (black, $\theta_\gamma = 7.9 \times 10^{-2}$), oil (brown, $\theta_\gamma = 1.3$) and glycerine (purple, $\theta_\gamma = 3.9$). Apart from mercury and silicone oil, these are also shown by LeBlond and Mainardi [50]. The silicone oil considered here is that used in the circular jump experiment of [55], with a viscosity about 20 times that of water. As mentioned in the text, the short wavelength cut-off for glycerine occurs at a larger wavelength than the capillary length, so that the cut-off occurs before reaching the capillary regime. In water, on the other hand, the cut-off wavelength is far shorter than the capillary length, so that we are well into the capillary regime before it occurs. Notice also that the imaginary part of the frequency is typically far smaller than the real part, and the cut-off occurs only shortly after that wave number whose real and imaginary parts are equal.

## 3.5   Weakly dissipative wave equation for surface waves

For the purpose of numerical simulations of analogue gravity experiments, our main concern is to take account of the high dissipation rate of capillary waves. We thus restrict ourselves to the weak bulk dissipative model represented by Eq. (19), i.e. we simply take the imaginary part of the frequency to be $\Gamma = 2\nu k^2$ (stronger dissipation is, for the purpose of describing experiments, beyond the scope of the present paper). When $\theta_\gamma$ is small (as in water), this provides a very good approximation to the effects of bulk viscosity over a wide range of wavelengths. Boundary friction is neglected, partly for simplicity as the expressions 15 for the corresponding damping rates are rather complicated. However, this approach can be justified by the relatively large dissipative lengths associated to boundary friction for long wavelengths as compared to the rather short dissipative lengths engendered by bulk viscosity at short wavelengths (see Fig. 3).

The wave equation we shall use is a straightforward generalization [11] of the Unruh model [5, 53, 62] in which an effective diffusive term is added to the time-derivative, i.e. $\partial_t \rightarrow \partial_t - 2\nu\partial_x^2$:

$$\left[\left(\partial_t - 2\nu\partial_x^2 + \partial_x u\right)\left(\partial_t - 2\nu\partial_x^2 + u\partial_x\right) + F^2\left(-i\partial_x\right)\right]\phi = 0\,, \tag{22}$$

---

[11] Note that the Unruh model is itself a dispersive generalization of the d'Alembertian equation in a $1 + 1$-dimensional spacetime with the metric $ds^2 = c^2\,dt^2 - (dx - v\,dt)^2$, where $c$ is a constant but $v$ is generally $x$-dependent. This leads to Eq. (22) with the co-moving frequency $F(k) = ck$.

where $u$ is the background flow velocity and $F(k)$ is the real part of the frequency in the rest frame of the fluid. This can be seen by assuming that $u$ is constant, and letting $\phi$ be the plane wave $\exp(ikx - i\omega t)$; then Eq. (22) implies

$$\omega - uk = \pm F(k), \qquad \Gamma = 2\nu k^2. \tag{23}$$

The first equation is equivalent to Eq. (14), while the second encodes weak dissipation in the bulk of the same form as Eqs. (19). Equation (22) can be solved numerically using finite difference methods; see Ref. [5] for an algorithm used to solve Eq. (22) with $\nu = 0$.

In an inhomogeneous background such as that provided by a flow over an obstacle, incoming linear waves are scattered into various outgoing waves by the inhomogeneities, but if the flow is stationary this process is restricted by the conservation of the lab frequency $\omega$. Thus, upon inspection of the dispersion relation, one can determine which outgoing waves a certain incident wave can potentially scatter into. In the adiabatic or "smoothly-varying" regime, the behavior of each wave is well-described by ray trajectories, which are solutions of Hamilton's equations [12]:

$$\frac{dx}{dt} = \frac{\partial\omega}{\partial k}, \qquad \frac{dk}{dt} = -\frac{\partial\omega}{\partial x}. \tag{24}$$

Typically there are several such rays for a given $\omega$, depending on the number of independent solutions of the dispersion relation. Non-adiabaticities in the variation of the flow will induce scattering between the various rays, whose wave vectors are not continuously connected and whose coupling is not described by Eqs. (24).

In the non-dissipative model (i.e., $\nu = 0$), Eq. (22) has a first integral called the *norm* [13], given by

$$(\phi, \phi) \equiv i \int_{-\infty}^{+\infty} \left[ \phi^\star \left( \partial_t + v\partial_x \right) \phi - \phi \left( \partial_t + v\partial_x \right) \phi^\star \right] dx, \tag{25}$$

or, when $u$ is constant,

$$(\phi, \phi) = \frac{1}{\pi} \int_{-\infty}^{+\infty} |F(k)| \left[ \left| \widetilde{\phi}_{\text{pos.}}(k) \right|^2 - \left| \widetilde{\phi}_{\text{neg.}}(k) \right|^2 \right] dk, \tag{26}$$

where $\widetilde{\phi}_{\text{pos.}}$ and $\widetilde{\phi}_{\text{neg.}}$ are the Fourier transforms of the positive- and negative-norm branches, respectively. While the norm is real, it is not positive definite; in particular, $(\phi^\star, \phi^\star) = -(\phi, \phi)$. In fact, it is straightforward to show that, for a plane wave or a wavepacket strongly peaked in Fourier space, the sign of the norm is the same as the sign of the co-moving frequency $\omega - uk$ (in regions where $u$ is constant). Moreover, the sign of the wave energy is equal to the sign of the product of the conserved frequency $\omega$ and the norm [53], and is thus invariant under complex conjugation. Finally, we note that when $\nu \neq 0$ but $\Gamma/\omega \ll 1$, $(\phi, \phi)$ will be approximately constant over the typical timescale $1/\omega$, and norm conservation will remain a relevant and useful concept: the scattering process itself will still be norm-preserving, but the norm will decrease in a trivial fashion due to damping of the waves as they propagate to and from the scattering region.

---

[12]It is possible to describe ray trajectories in dissipative media using generalized versions of Hamilton's equations; see e.g. Ref. [63].

[13]This is closely related to conservation of wave action [64], to which it becomes equivalent in the adiabatic limit of geometrical optics; see also Ref. [22].

# 4 Applications to analogue gravity

In this section we consider several experiments with surface waves which are relevant to analogue gravity. We apply the considerations of previous sections to see how and to what extent they are affected by viscous dissipation, and where appropriate we indicate how the experimental setup (particularly in water) might be optimized in this light.

## 4.1 Hawking effect

Since the seminal paper by Unruh [1] first established the field, the main focus of the analogue gravity program has been on the analogue Hawking effect (see also Refs. [19, 21, 24] for experiments with water waves and [10] for observations in BEC). This is principally an example of *anomalous scattering*, which can be described as follows. Conservation of the total energy of linear waves imposes a constraint on the normalized scattering amplitudes, called the *unitarity relation*, such as $|R|^2 + |T|^2 = 1$ relating reflection and transmission coefficients. Anomalous scattering is the corresponding situation when one or more of the waves involved in the scattering process has negative energy, and thus counts negatively towards the unitarity relation. If a positive-energy incident wave partially scatters into negative-energy outgoing waves, then, in order for energy conservation to be respected, the positive-energy outgoing waves must between them possess an energy which is *greater* than the energy of the incident wave. Anomalous scattering is thus also a process of amplification [65], either of a classical incoming wave, or (as in the original scenario described by Hawking) of quantum vacuum fluctuations. [14] A necessary condition for anomalous scattering, then, is the existence of negative-energy waves. As seen in the phase diagram in the right panel of Fig. 2, this requires $|U| = |u/u_\gamma| > 1$ in at least one of the asymptotic regions.

An example dispersion relation on a flow with $|U| > 1$ is shown in the left panel of Figure 6. We focus on gravity waves, since in the relevant frequency range $|\omega| < \omega_{\max}$ the capillary branch retreats quickly to large $k$ for $|U| \gtrsim 1$; and we focus on counter-propagating waves, since only this branch is blocked and the co-propagating branch is largely decoupled from it [66]. We also assume that, while $|u| > u_\gamma$, it is not larger than the low-$k$ wave speed [15] $\sqrt{gh}$, allowing us to have both positive- and negative-energy waves at reasonably long wavelengths. The various branches of the dispersion relation are shown in different colors; there is one long-wavelength mode (in blue) and two short-wavelength modes (in red and yellow), the shortest wavelength (in yellow) having negative energy. The blue and red branches are converted into each other at a turning point in accordance with Hamilton's equations (24). Examples of the ray trajectories are shown in the right panel of Fig. 6 for a flow over a localized obstacle, and thus containing both a black hole and a white hole horizon.

---

[14]In its original gravitational context, the Hawking effect occurs because the outgoing modes which are trapped by the horizon have negative norm inside the black hole, so quantum fluctuations in the vicinity of the horizon can be "amplified" into outgoing particles, one of positive energy escaping to infinity, the other of negative energy falling into the black hole.

[15]It could be argued that the black (or white) hole analogy is appropriate *only* when $|u|$ crosses $\sqrt{gh}$, since this would ensure that *all* incident waves are blocked and predicts an anomalous scattering coefficient $|\beta|^2 \propto 1/\omega$ in the low-frequency regime, much as for the Planck spectrum. We do not enforce this condition here in order to conform to experimental setups that have already been realised [21, 23, 24]. Indeed, focusing as we do here on gravity waves, we must have $|u| < \sqrt{gh}$ in at least one asymptotic region in order for positive-energy waves to be present, so our results will be relevant there. Moreover, the modes in a region where $|u| > \sqrt{gh}$ tend to be long-wavelength, and of less relevance when considering the dissipative effects of viscosity.

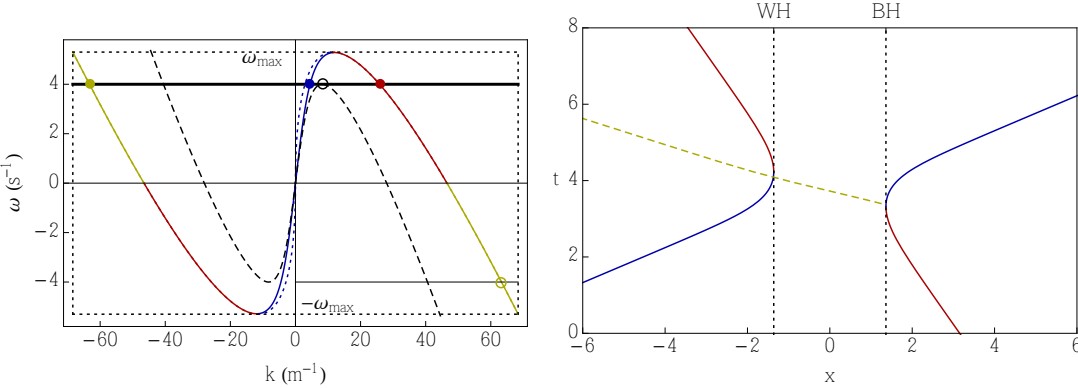

Figure 6: Wave blocking and the analogue Hawking effect. On the left is plotted (in color) the dispersion relation of counter-propagating gravity waves on water for $u = -2u_\gamma = -46.2\,\mathrm{cm/s}$ and a water depth $h = 25\,\mathrm{cm}$; the dotted curve corresponds to the deep limit $h \to \infty$. The different colors correspond to the different branches of the dispersion relation, $k_j(\omega)$, that exist for $0 < \omega < \omega_{\mathrm{max}}$. As an example, the thick black line indicates $\omega = 4\,\mathrm{Hz}$, and the colored dots show the corresponding wave vector solutions. (We note in passing that, since the dispersion relation is an odd function of $k$, one can find the negative-$k$ solutions by focusing on $k > 0$ and adding a horizontal line at $-\omega$, as shown by the thin black line.) If $|U|$ increases in the direction of propagation, the "local" dispersion relation dips, reaching the dashed curve at $u \approx -2.5\,u_\gamma$. There, the horizontal line is tangent to the curve; this corresponds to a turning point in the characteristics of the wave equation, examples of which are shown in the right panel for a flow over a localized obstacle and are solutions of Hamilton's equations (24). Horizons then necessarily occur in black hole–white hole pairs, and there are two types of wave blocking: at the black hole horizon, the wave number is reduced so that the red branch is converted into the blue branch; and at the white hole horizon, the inverse process occurs. Both are adiabatic effects described by geometrical optics, but there can also be some non-adiabatic scattering into the yellow branch, whose energy is negative and whose production thus necessitates an increase in the energy of the positive-energy wave.

At the black hole horizon, $u'(x) > 0$ and the second of Eqs. (24) gives $\dot{k} < 0$, so the red branch of the dispersion relation is converted into the blue branch. The more commonly studied case in water wave physics is the white hole horizon, at which the inverse of this process occurs [21, 24]. In both cases, however, non-adiabaticity of the variation of the flow profile results in some production of the negative-energy wave on the yellow branch; this is the analogue of the Hawking effect.

In Figure 7 are plotted the dissipative lengths for the waves relevant to the analogue Hawking effect. In the left panel are shown, for a fixed flow velocity of $2\,u_\gamma$, the dissipative lengths for each of the branches of the dispersion relation, both with (solid curves) and without (dashed curves) taking boundary friction into account (assuming a channel width of $39\,\mathrm{cm}$, as in the experiments of [23, 24, 67]). It is clear that the short-wavelength (yellow) branch is most prone to dissipation, even when boundary friction is included. We focus on this short-wavelength branch in the right panel, where now we plot its dissipative length as a function of the flow velocity. [16] Its maximum and minimum dissipative lengths are plotted both with

---

[16]Note that the dissipative length of the short-wavelength modes does not generally depend on the depth

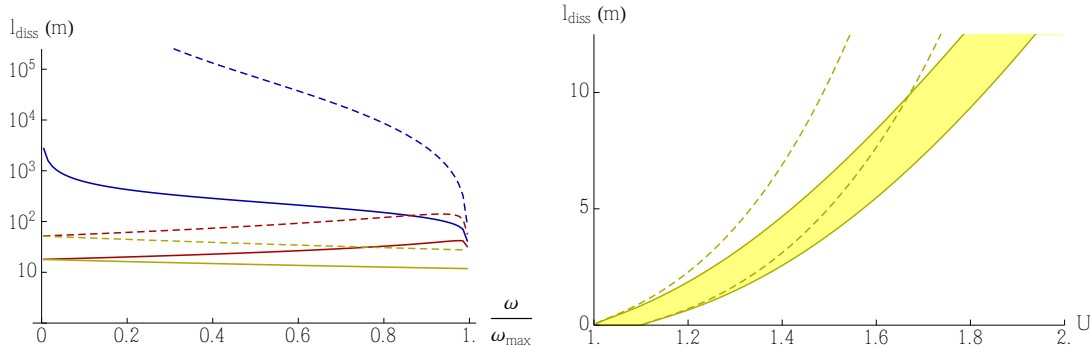

Figure 7: Dissipative lengths appropriate to the analogue Hawking effect in water. On the left are shown the dissipative lengths of the three branches of the dispersion relation shown in Fig. 6, with the colors corresponding to the colors used there. The dashed curves correspond to bulk dissipation only, whereas the solid curves include loss due to boundary effects, for a depth $h = 25$ cm and a channel width $b = 39$ cm. Although we have fixed the flow velocity at $2\,u_\gamma$, changing $u$ does not qualitatively change the plot until capillary effects become important (which occurs around $u \approx 1.1\,u_\gamma$). In particular, we note that the branch shown in yellow, with the highest wave vectors, generally has the shortest dissipative length. On the right are plotted, as a function of the flow velocity, the range of dissipative lengths corresponding to the yellow branch. Again, the solid curves which are filled in correspond to a finite channel width of $39$ cm, whereas the unfilled dashed curves show the envelope obtained due to bulk dissipation alone, i.e. in the limit where boundary friction vanishes.

(solid curves) and without (dashed curves) boundary friction, where the former has been filled in for clarity since any point on the yellow branch must lie in between these limiting values. We note that the dissipative length is always well above $10$ m when $|U| \gtrsim 2$, so for such flow velocities viscous dissipation is expected to be negligible for those waves relevant to the analogue Hawking effect.

## 4.2 Black hole laser effect

We have seen that anomalous scattering induces amplification by coupling waves of opposite energy. Here we turn to a related situation in which this amplification is exponential in time, thus rendering the system dynamically unstable. [17] This is achieved by the presence of trapped modes, caught between two turning points so that they cannot escape to infinity except through non-adiabatic coupling to free modes. If the free modes to which they couple have opposite energy, the trapped mode becomes self-amplifying through successive anomalous scattering events; see the ray trajectories, solutions of Hamilton's equations (24), in Figure 8. This is known as the black hole laser effect [70] (see Ref. [9] for an experimental implementation in BEC). It can be described by a number of discrete [18] unstable modes of complex frequencies

---

$h$, since for typical depths this affects only the long-wavelength part of the dispersion relation; see the dotted curve in the left panel of Fig. 6.

[17]In practice the dynamical instabilities are regularized by nonlinear effects, not included in the linear theory used in this paper. See [68, 69] for a study of nonlinear effects in BEC black hole lasers.

[18]This is a notable difference with respect to the instabilities induced in the rotating conducting cylinder (see Sec. 2.3), which form part of the continuous spectrum.

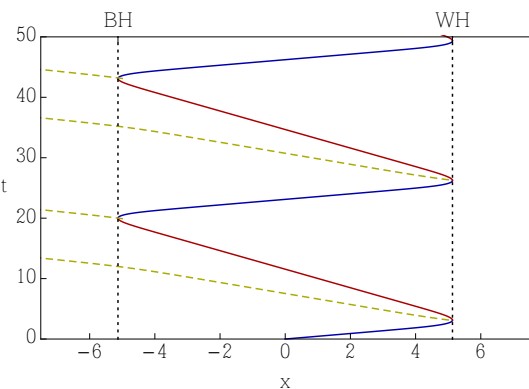

Figure 8: Characteristics in a (gravity-regime) black hole laser. The black hole–white hole horizon pair is arranged in such a way that the outgoing ray after blocking at one horizon is subsequently blocked at the other (indicated here by the blue and red curves). The ray is unable to escape from the interior region; this is what is meant by a trapped mode. Moreover, since it couples to a free mode of opposite energy (the dashed yellow curves), it is continuously amplified at each blocking, making the system dynamically unstable.

which grow exponentially in time [71, 72]. At late time, the behavior is governed by the most unstable of these modes, and the amplitude increases exponentially at a well-defined rate.

For surface waves, there are two possible configurations of the background flow leading to the existence of trapped modes and to the black hole laser effect. These can be termed the gravity- and capillary-regime black hole lasers, since their trapped modes lie respectively on the gravity and capillary branches of the dispersion relation. The gravity-regime black hole laser is particularly simple to describe, since (much as for the analogue Hawking effect) we can neglect the capillary branch altogether and focus solely on gravity waves. Consider a flow consisting of two asymptotic "outside" regions and an interior region where the flow is slower (an example is shown in the left column of Figure 9). Such a flow might be achieved by means of a trough, where the fluid is deeper over a finite region than it is in the exterior regions. There will be certain positive-energy modes whose rays are trapped in the interior region; furthermore, so long as $|U| > 1$ is large enough outside, these modes will couple to negative-energy modes which can escape to infinity. This is precisely the scenario that can engender dynamical instabilities and the onset of the black hole laser effect.

The capillary-regime black hole laser occurs when the flow is faster in the interior region than it is in the asymptotic regions, as might be engendered by a bump on the bottom of a flume so that the fluid is shallower in the interior region (an example is shown in the right column of Fig. 9). Negative-energy waves must therefore exist in the interior region, where we must have $|U| > 1$. Some of these negative-energy waves will form trapped modes, and these will couple to positive-energy waves in the asymptotic regions. The rather complicated dispersion relation induces a richer phenomenology than in the gravity-regime black hole laser, but the essential points remain the same.

The most crucial difference between the two regimes concerns the role of dissipation: since the trapped modes in the capillary-regime black hole laser occur at significantly larger wave numbers than their counterparts in the gravity-regime, they will be much more vulnerable to viscous damping. It is thus of interest to consider optimization of the capillary-regime black hole laser, to see whether it is physically realisable or if dissipation excludes this possibility.

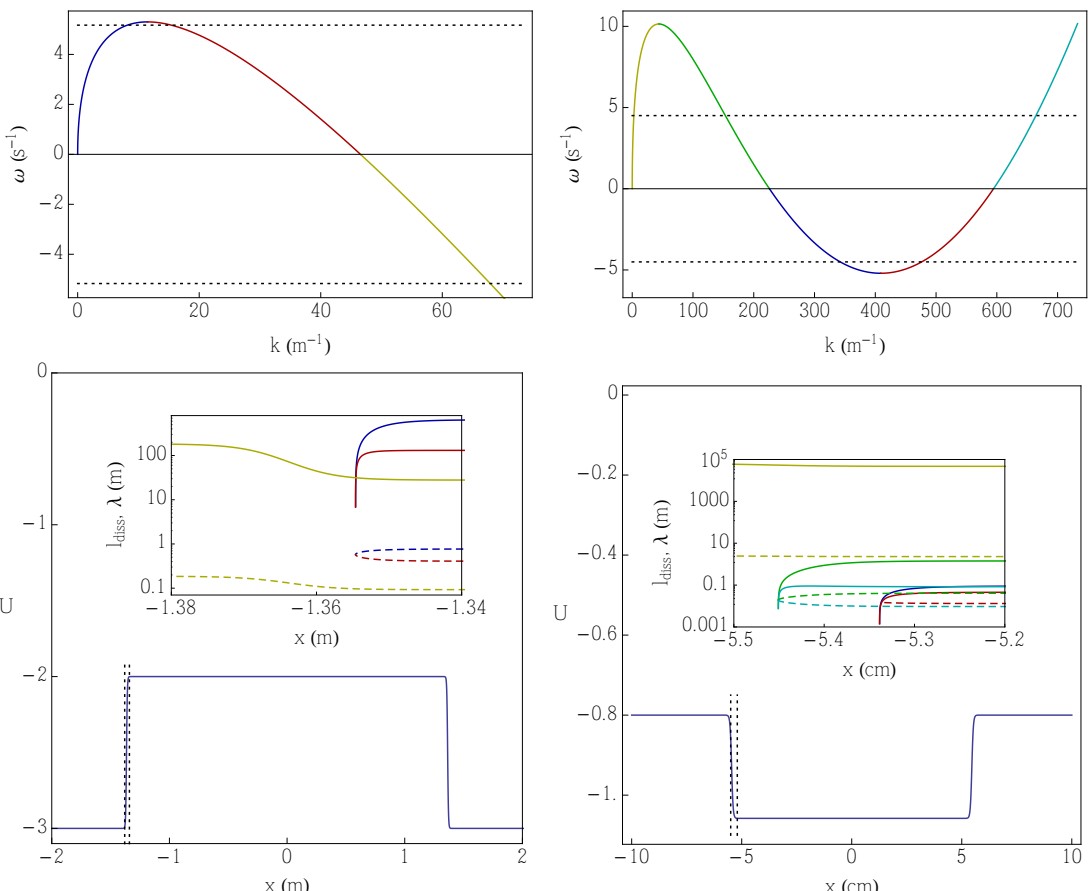

Figure 9: Flow profiles for black hole laser setups in the gravity regime (left column) and the capillary regime (right column). The top row shows the corresponding dispersion relations in the inner cavity region, with the various branches colored differently for clarity. The horizontal dotted lines show $\pm\omega_0 \equiv \pm\omega(k_0)$, where $k_0$ is the central wave number of the initial Gaussian wavepacket used in the numerical simulations. The bottom row shows the flow velocity profiles used in each case. In the insets are plotted (in a logarithmic scale) the dissipative length and the wavelength as a function of position, where we have zoomed in on the transition region delimited by vertical dotted lines in the flow profile. For the purpose of having a clear numerical signal that is amplified fairly quickly, the derivative $v'(x)$ is unrealistically large in these regions, especially for the capillary-regime example on the right. The curves are colored according to which branch of the dispersion relation they belong to, with the dissipative length shown by the solid curves and the wavelength shown by the dashed curves.

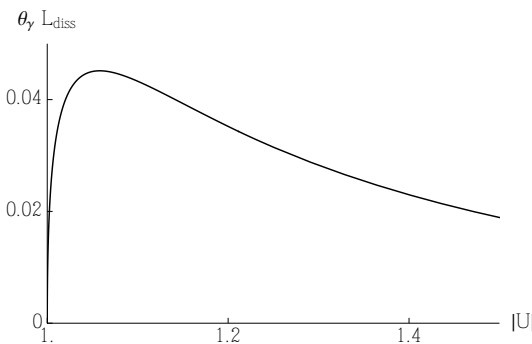

Figure 10: Optimizing the capillary-regime black hole laser. Plotted here is the maximum dissipative length occurring on the high-$k$ negative-energy branch (shown in red in the upper right panel of Fig. 9), which is that most relevant to the trapped mode. There is a clear maximum of $\theta_\gamma L_{\mathrm{diss}} = 0.045$ at $|U| = 1.058$ (in water, this corresponds to an optimum dissipative length of 7.8 cm at a flow velocity of 24.5 cm/s).

To this end, we note that the trapped modes lie on the negative-energy part of the dispersion relation (that shown in blue and red in the upper right panel of Fig. 9), and that the branch most susceptible to dissipative effects is that with higher $k$ (i.e. the red branch). We thus plot, in Figure 10, the maximum dissipative length occurring on this branch as a function of the interior flow velocity. There is a clear maximum at $|U| = 1.058$, where $\theta_\gamma L_{\mathrm{diss}} = 0.045$; in water, this corresponds to an interior flow velocity of 24.5 cm/s and a maximum dissipative length of 7.8 cm.

    We performed numerical simulations of Eq. (22) using the finite difference method described in Ref. [5]. The flow velocity profile $U(X)$ was as given in the lower panels of Fig. 9 for the gravity-regime (left column) and capillary-regime (right column) black hole lasers, respectively. We assumed the deep water limit $F^2(K) = \left(|K| + |K|^3\right)/2$, though this can easily be generalized to the case of finite depth (see Eq. (12)). The initial conditions for the field $\phi$ are that it lies on the counter-propagating branch (this determines the "canonical momentum" $\pi = (\partial_t + u\partial_x)\phi$ once $\phi$ is known), and that it is a normalized Gaussian wave packet entirely contained within the interior region and having a wave vector which is "trapped" in the sense described above and illustrated in Fig. 8. The initial wave vector lies on the blue-colored branch in the top row of Fig. 9 (both for the gravity-regime and capillary-regime black hole lasers), and has the frequency marked by the horizontal dotted line.

    Although it is strictly correct only when $u$ is constant, Eq. (26) gives a very good approximation to the norm of the solution at any particular time, and is useful in that (assuming that the contribution from the co-propagating branch is negligible) it allows us to separate the positive- and negative-norm components as simply the integral over positive and negative $k$, respectively. In Figure 11 are plotted some numerical results showing the growth of the negative-norm component. The capillary-regime black hole laser is chosen to be close to optimal (in the sense described above), with $|U| = 1.058$ in the interior region and the inter-horizon distance being about 1.5 times the maximum dissipative length on the high-$k$ negative-energy branch. What is especially clear is that, while dissipation does have a cumulative effect on the instability of the gravity-regime black hole laser [19], its effect on the

---

[19]Since we use Eq. (22) to numerically model the black hole laser, dissipation due to friction at the boundaries

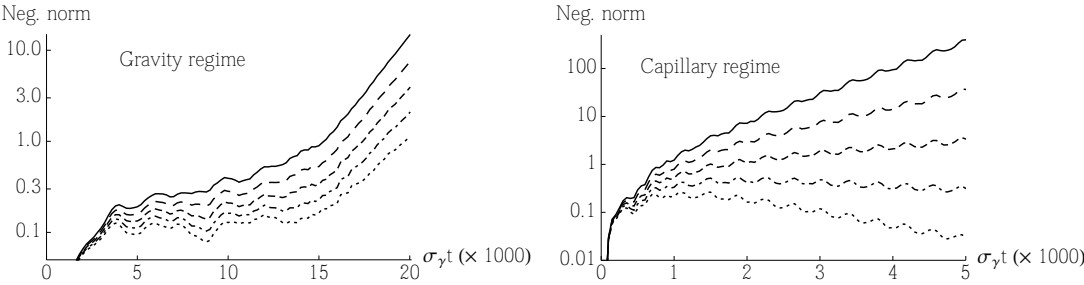

Figure 11: The negative-norm component as a function of time in both the gravity-regime (left) and the capillary-regime (right) black hole lasers. The initial conditions consist of a purely trapped component, which is taken to be Gaussian in space and to have positive-norm set initially to 1. The various curves correspond to different values of the adimensionalized viscosity $\theta_\gamma$: in the left panel these are 0 (solid), $1 \times 10^{-3}$ (long dashed), $2 \times 10^{-3}$ (short dashed), $3 \times 10^{-3}$ (dot-dashed) and $4 \times 10^{-3}$ (dotted); while in the right panel they are an order of magnitude smaller, being 0 (solid), $1 \times 10^{-4}$ (long dashed), $2 \times 10^{-4}$ (short dashed), $3 \times 10^{-4}$ (dot-dashed) and $4 \times 10^{-4}$ (dotted). Note that, after a rather complicated transient evolution, the growth becomes exponential at large $t$, indicating the late-time dominance of the most unstable mode. The rate of this growth decreases roughly linearly with $\theta_\gamma$, until $\theta_\gamma$ becomes large enough for the growth to be completely suppressed, as for the last two curves in the right panel. Since in water $\theta_\gamma = 1.58 \times 10^{-3}$, it is clear that the gravity-regime black hole laser is not much affected by viscosity, whereas the capillary-regime black hole laser (taken here to be close to 'optimal', with $|U| = 1.058$ in the interior region and the inter-horizon distance being about 1.5 times the maximum dissipative length on the high-$k$ negative-energy branch) is killed off by this value.

capillary-regime configuration is much more drastic, where it kills off the instability at a relatively small dissipation rate, even though it is close to 'optimal' in the sense of Fig. 10. We thus conclude that, in water, the capillary-regime black hole laser is not physically feasible.

## 4.3   Wormhole traversal

We have already seen (in studying the Hawking effect and the gravity-regime black hole laser) that gravity waves are sufficient for the realization of turning points, where the ray trajectories change direction. If we exploit the full dispersion relation of gravity-capillary waves, their propagation becomes even richer through the appearance of "double bouncing", in which the rays encounter a second turning point and return to their initial direction of propagation, even as the wave number varies monotonically throughout. This double bouncing phenomenon was observed in [73], and studied numerically in [20,74]. Thus, while gravity waves are forbidden from passing the first turning point, the possibility of reaching the capillary regime gives the wave a way past it, allowing it to enter an apparent white hole and/or escape from an apparent black hole. For a flow in which both are present, the wave is thus able to traverse the previously forbidden region between the two. It is this that has been termed the analogue of "reverse" wormhole traversal [75], i.e. propagation in the "forbidden" direction from the white hole to the black hole. It has recently been observed experimentally in [67].

---

has been neglected. This would likely decrease the growth rate for the gravity-regime black hole laser, but probably not very significantly since the dissipative length typically remains large (as we saw in Figs. 3 and 7).

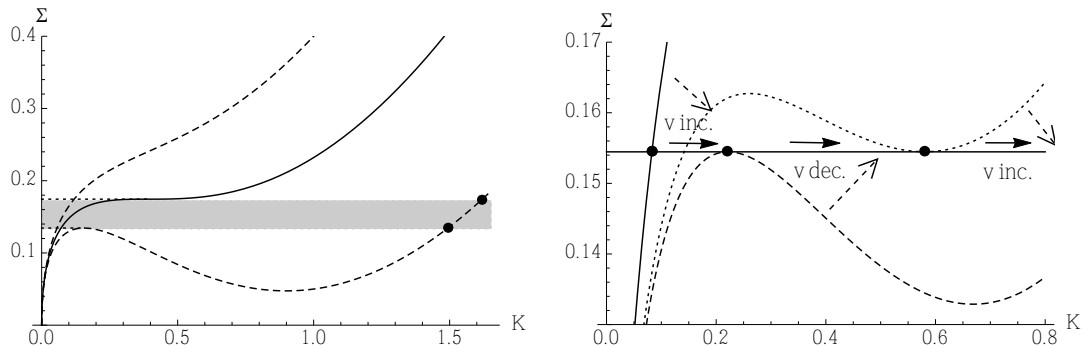

Figure 12: Dispersion diagrams indicating the double bouncing phenomenon upon entering an analogue wormhole. On the left, the solid curve shows the dispersion relation at the critical flow velocity $|U| = 0.768$, which has a point of inflexion. The dashed curves show examples of the dispersion profile for typical flow velocities outside ($|U| < 0.768$) and inside ($|U| > 0.768$) the wormhole. Note that the latter has local extrema while the former does not. The band lying between the local maximum of the dispersion relation inside the wormhole and the critical frequency at which the point of inflexion occurs is shown in grey; it is the frequencies within this band that will experience double bouncing. The right panel illustrates this, with a zoomed-in version of the dispersion profile. The near-vertical solid curve corresponds to the dispersion relation in the asymptotic region outside the wormhole, while the horizontal solid line represents a frequency for which double bouncing can occur. Consider an incident wave at this frequency, whose wave vector is given by the intersection of the two solid lines (i.e. at the left-most black disk). As it begins to enter the wormhole, the flow velocity increases, and the dispersion diagram tilts downward. The wave vector thus increases until it coincides with the maximum of the local dispersion relation (i.e. the second black disk at the extremum of the dashed curve). At this point, the wave turns around, moving back towards lower flow velocities so that the dispersion relation tilts back upwards. However, its wave vector continues to increase, until once again it reaches an extremum, this time a minimum, of the dispersion relation (i.e. the right-most black disk at the extremum of the dotted curve). It turns around once again, moving towards larger flow velocities and all the while continuing to increase its wave vector. It encounters no more extrema, fully entering the wormhole and settling on a high wave vector on the right side of the grey band shown in the left panel.

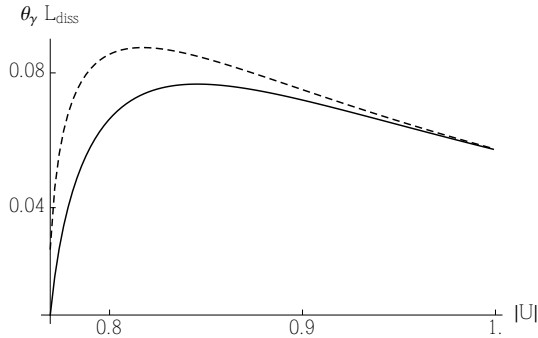

Figure 13: Adimensionalized dissipative length for solutions in the interior region of an analogue wormhole, as a function of the interior flow velocity. These correspond to the minimum (solid curve) and maximum (dashed curve) frequencies at which double bouncing occurs; that is, they correspond to the wave vectors at either end of the large-$k$ strip whose frequencies lie in the shaded band in the left panel of Fig. 12. The lower limit is at $|U| = 0.768$, since this is the minimum interior flow velocity required for double bouncing. We have taken the upper limit at $|U| = 1$, since above this value negative-energy waves exist in the interior region, and black hole lasing can occur; however, the curves continued to decrease smoothly for $|U| > 1$. Surprisingly, the dissipative length is greater for the larger wave vectors (near the maximum of the double bouncing frequency band), this being due to their larger group velocity allowing them to propagate further before being dissipated. This takes its largest value of $\theta_\gamma L_{\text{diss}} \approx 0.0875$ when $|U| \approx 0.817$ (for water, this corresponds to a maximum dissipative length of 15.1 cm at an interior flow velocity of 18.9 cm/s). Note that the dissipative length decreases rather slowly with increasing $|U|$, so we can afford to let the $|U|$ be a bit larger than 0.817.

In order for double bouncing to occur, it is clear that a turning point must be present, and hence that the flow speed in the inter-horizon region must be greater than the minimum group velocity $U_c$ on a static fluid. For simplicity, we assume that the flow is less than $U_c$ in the exterior regions. Double bouncing then occurs for frequencies between a universal maximum ($\Sigma_{\text{max}} \equiv \omega_{\text{max}}/\sigma_\gamma = 0.174$; in water, this gives $\omega_{\text{max}} = 14.8$ Hz or $T_{\text{min}} = 0.425$ s) and a minimum that depends on the inter-horizon flow velocity; see the shaded band in the left panel of Figure 12. It is clear from this figure that the wave propagating in the interior region lies in a small window of both frequency and wave number well inside the capillary regime, so that viscous daming will be particularly important there. In Figure 13 are plotted, as functions of the interior flow velocity, the adimensionalized dissipative lengths of the waves at either limit of this window (indicated by black disks in the left panel of Fig. 12). There is a clear maximum at $|U| \approx 0.817$, where $\theta_\gamma L_{\text{diss}} \approx 0.0875$; in water, this corresponds to $u \approx 19$ cm/s, $l_{\text{diss}} \approx 15$ cm. Although this is the maximum, the dissipative length decreases rather slowly as the interior flow velocity is increased, so that there is some freedom for $|U|$ to be a bit larger than 0.817.

In Figure 14 are shown some numerical results of analogue wormhole traversal, using again the finite difference method described in Ref. [5] to solve Eq. (22). The flow velocity $|U|$ is chosen to be fairly close to optimal, varying from 0.7 outside to 0.85 inside the wormhole. The initial conditions for $\phi$ are that it is a Gaussian wave packet on the counter-propagating branch, lying entirely to the left of the analogue wormhole. In all simulations, the central

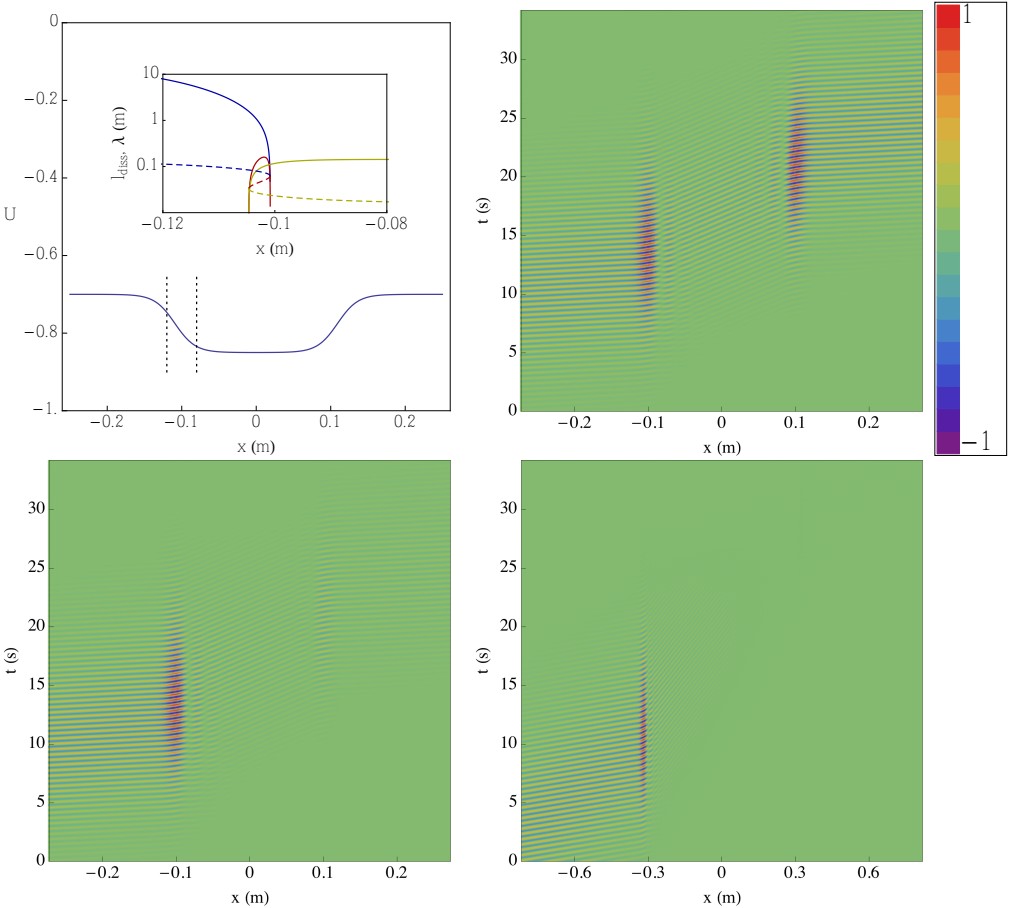

Figure 14: Propagation through an analogue wormhole. The upper left panel shows the flow velocity profile used in the numerical simulation, in units of the minimum phase velocity $u_\gamma$. Throughout, the water is assumed to be infinitely deep ($h \to \infty$). The inset of the upper left panel shows, in logarithmic scale, the evolution of the wavelength (dashed curves) and the dissipative length for water (solid curves) with the flow, in the "double bouncing region" delimited on the main plot by the dotted lines. In blue we show the evolution of these lengths before the first turning point is reached, red shows how they behave during the backward propagation between the two turning points, and yellow corresponds to the propagation away from the second turning point and into the interior region. The remaining panels show space-time diagrams of the wave traversing the wormhole, in which the wave is normalized such that its maximum and minimum values are 1 and $-1$, respectively; the corresponding color legend is shown on the right. The upper right panel is in the absence of any dissipative effects (thus corresponding to the non-dissipative simulations performed in [75]). It is clear that, in the inner region, the wavelength is considerably shorter and the total wave speed considerably slower than in the outer regions. The lower left panel shows the same wave propagation with viscosity set to its value for water (i.e. $\nu = 10^{-6}\mathrm{m}^2/\mathrm{s}$), so that the inter-horizon distance is about 1.5 times the interior dissipative length and the outgoing wave is strongly damped but still visible (this scenario closely corresponds to the recent observations in [67]). In the lower right panel, the viscosity remains that of water but the obstacle has been made three times longer, so that the inter-horizon distance is now about 4.5 dissipative lengths and no outgoing wave is visible (this corresponds to the observations in [73]).

frequency of the wavepacket is fixed at $\omega = 14\,\mathrm{Hz}$, with corresponding period $T = 0.45\,\mathrm{s}$. The width of the wavepacket is chosen such that, in Fourier space, it is almost entirely contained within the double bouncing "window" (the shaded region in the left panel of Fig. 12), for otherwise a part of the wavepacket would be seen to split off, being either reflected or directly transmitted. In the first two simulations, the length of the wormhole (i.e. the distance between the black- and white-hole horizons) is set to about $20\,\mathrm{cm}$. In the first simulation, dissipation is set to zero, so that we see pure, unimpeded propagation across the wormhole. Note the increase in amplitude in the near-horizon regions: this is due to the piling-up of the wave as its group velocity vanishes at the two turning points. [20] In the second simulation, the viscosity is set to that of water (i.e. $\theta_\gamma = 1.58 \times 10^{-3}$, or $\nu = 10^{-6}\,\mathrm{m}^2/\mathrm{s}$). Then, the dissipative length of the chosen wavepacket is about $14\,\mathrm{cm}$ in the interior region, and the length of the wormhole is about 1.5 dissipative lengths. We see that, while the amplitude of the transmitted wave is significantly reduced, it remains visible. In the third simulation, the viscosity is again set to that of water, but the length of the wormhole is increased to $60\,\mathrm{cm}$. This is now about 4.5 dissipative lengths, and the transmitted wave is no longer visible on a linear scale.

## 4.4   Double bouncing

The double bouncing behavior of the characteristics is a crucial element in our description of the above-described setup as an analogue wormhole: there is an initial turning point due to the long-wavelength dispersion, akin to the horizons we are accustomed to; and it is only by going into the short-wavelength regime that new dispersive behavior can be exploited, and the initial turning point crossed. But the double bouncing itself is typically hard to see clearly because the two turning points involved are very close together. For water, this seems to be unavoidable if we require the wave to cross the wormhole without being completely dissipated (as in the experiments of [67]). However, if we focus on the double bounce alone, at the white hole side of the wormhole only, and allow the wave to be completely dissipated afterwards, it may be possible to separate the turning points to such an extent that the double bouncing becomes resolvable (as in the experiments of [73] and the numerical simulations of [74]).

   We can perform a rough optimization of the resolution of the double bounce as follows. Firstly, even in the absence of dissipation it is clear that many setups are undesirable: for, if the wave packet is too wide, it will obscure the double bounce. It is thus necessary to use wave packets that are significantly narrower in position space than the distance between the two turning points. But there is a also a limit to this narrowness, for the turning points themselves are frequency-dependent, so that there is a "spread" to the turning points by virtue of the spread of the wave packet in Fourier space.

   To mathematize this, let us use the label $k_{\mathrm{t1}}$ to denote the central wave number of the wave packet at the first turning point. The flow velocity at the turning point is a function of $k_{\mathrm{t1}}$, and in fact it is just the negative of the group velocity in the rest frame of the fluid: $u_{\mathrm{t1}}(k_{\mathrm{t1}}) = -\omega_0'(k_{\mathrm{t1}})$. Then the spread in the turning point velocities is given by $\Delta u_{\mathrm{t1}} \approx |\omega_0''(k_{\mathrm{t1}})|\,\Delta k_{\mathrm{t1}}^{\mathrm{wp}}$, where $\Delta k_{\mathrm{t1}}^{\mathrm{wp}}$ is the spread of the wave packet in Fourier space when it is at the first turning point. Moreover, if for the sake of simplicity we assume that the flow velocity profile is linear in $x$, i.e. $u(x) = -u_0 - \alpha x$ where $u_0$ and $\alpha$ are both positive, then

---

[20]We assume that the amplitude is small enough that the increase near the turning points does not lead to any nonlinear behavior.

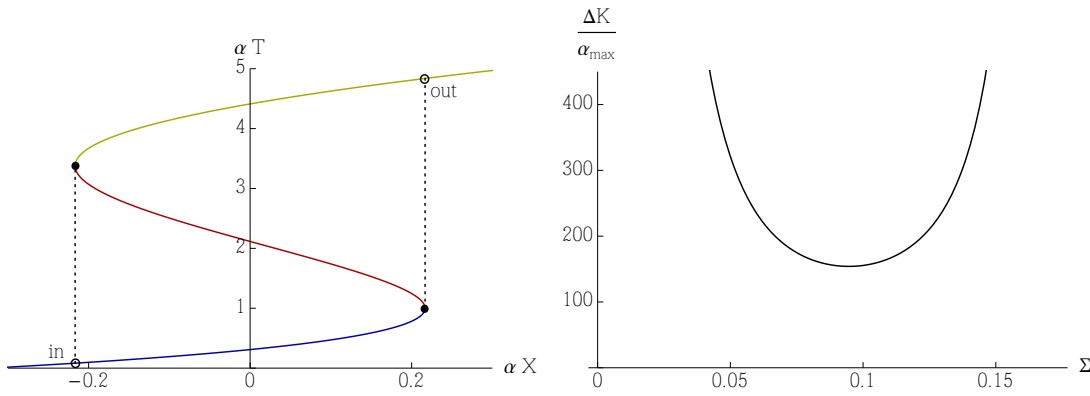

Figure 15: Optimizing the visibility of the double bouncing. On the left is plotted a wave characteristic clearly exhibiting double bouncing. The two turning points are indicated by black disks, while the circles labelled 'in' and 'out' are the endpoints at which the initial and final wave numbers are evaluated, for insertion into expression (30) for the overall dissipation factor. On the right is plotted the (adimensionalized) exponent of this dissipation factor, which has a clear minimum at $\Sigma \sim 0.09$ (corresponding to $\omega \approx 8.0\,\text{Hz}$ in water, or $T \approx 0.78\,\text{s}$).

the spread in the turning point positions is

$$
\Delta x_{\text{t1}}^{\text{tp}} \approx \frac{|\omega_0''(k_{\text{t1}})|}{\alpha} \Delta k_{\text{t1}}^{\text{wp}} . \tag{27}
$$

In order for the double bouncing to be resolvable, this should be significantly smaller than the distance between the two turning points, $\Delta u_{\text{tp}}(k_{\text{t1}})/\alpha$, where the difference in the turning point velocities $\Delta u_{\text{tp}}$ is itself a function of $k_{\text{t1}}$. However, the width of the wavepacket itself, $\Delta x_{\text{t1}}^{\text{wp}} \approx 1/\Delta k_{\text{t1}}^{\text{wp}}$, should also be considerably smaller than the distance between the turning points. These two conditions combined give

$$
\frac{\alpha}{\Delta u_{\text{tp}}} \ll \Delta k_{\text{t1}}^{\text{wp}} \ll \frac{\Delta u_{\text{tp}}}{|\omega_0''(k_{\text{t1}})|} , \tag{28}
$$

from which we immediately find an upper limit for the slope $\alpha$:

$$
\alpha \ll \frac{(\Delta u_{\text{tp}})^2}{|\omega_0''(k_{\text{t1}})|} \equiv \alpha_{\text{max}} . \tag{29}
$$

Conditions (28) and (29) come from the requirement that the double bouncing be well-resolved. Our final task is to ensure that dissipation does not obscure it entirely. Indeed, one can always resolve the double bounce by making $\alpha$ small enough, but the danger then is that the wave will be dissipated before the double bounce is completed. We should therefore include the effects of bulk viscosity. Since we assume a linear velocity profile, the evolution of the wave number is given straightforwardly by the second of Hamilton's equations: $\dot{k} = \alpha k$, or $k(t) = k_0 e^{\alpha t}$. The dissipative rate is simply $\Gamma(t) = 2\nu k^2(t) = 2\nu k_0^2 e^{2\alpha t}$, and the dissipative factor is

$$
\exp\left(-\int_{t_{\text{in}}}^{t_{\text{out}}} \Gamma(t')dt'\right) \qquad \text{where} \qquad \int_{t_{\text{in}}}^{t_{\text{out}}} \Gamma(t')dt' = \frac{\nu}{\alpha}k_0^2\left(e^{2\alpha t_{\text{out}}} - e^{2\alpha t_{\text{in}}}\right) = \frac{\nu}{\alpha}\Delta k , \tag{30}
$$

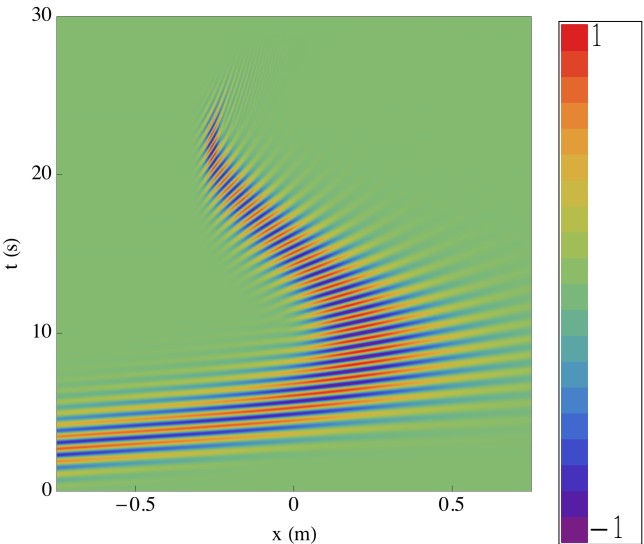

Figure 16: Numerical realisation of the optimized double bouncing in water. In real units, the frequency of the wave is $\omega = 8.02\,\mathrm{Hz}$ (equivalently, the period is $T = 0.783\,\mathrm{s}$), while the derivative of the flow velocity is $du/dx = -0.184\,\mathrm{s}^{-1}$, with $u = -30.7\,\mathrm{cm/s}$ at the first turning point and $-20.7\,\mathrm{cm/s}$ at the second turning point. In the experiment of [73], the largest period $T = 0.67\,\mathrm{s}$ while the slope is about $-0.1\,\mathrm{s}^{-1}$, not too far from the parameters used here. The wave is normalized such that its maximum and minimum values are 1 and $-1$, respectively; the corresponding color legend is shown on the right.

where $\Delta k \equiv k\,(t_{\mathrm{out}}) - k\,(t_{\mathrm{in}})$. For the purposes of optimization, we can use the maximum value of $\alpha$ allowed by (29), and choose the turning points as the initial and final positions at which $k(t)$ should be evaluated. This gives a curve for the exponent of the dissipative factor as a function of frequency that possesses a well-defined minimum, and it is this we choose as the "optimized" frequency. It is shown in the right panel of Figure 15, and gives an optimum frequency of $\Sigma = 0.095$, which in water is equivalent to $\omega = 8.0\,\mathrm{Hz}$ or $T = 0.78\,\mathrm{s}$. This is to be compared with the parameters used in the experiments of [73], where the largest period $T$ was $0.67\,\mathrm{s}$, not too far from the 'optimal' value. It can also be compared with the numerical simulations of [74], which used a period of $T = 0.53\,\mathrm{s}$, a bit further from the value we have used.

Solving Eq. (22) numerically using the finite difference method of Ref. [5], we have produced a space-time diagram for a wave packet at the optimum frequency extracted from the right panel of Fig. 15. The slope $\alpha = \alpha_{\mathrm{max}}/10$, which in water corresponds to $du/dx = -0.18\,\mathrm{s}^{-1}$. Again, this is to be compared with the parameters used in [73], where the slope was around $-0.1\,\mathrm{s}^{-1}$, almost a factor of 2 smaller than the value used here. In [74], the simulation closest to ours (see Fig. 10 of that work) used $du/dx = -0.033\,\mathrm{s}$, a factor of 6 smaller than ours. To the left of the turning points, $u$ becomes constant at $-18.4\,\mathrm{cm}$ for the convenience of preparing the initial wave packet there. It is given a width such that the inequalities of 28 are realized; specifically, it has a wavelength of $63.5\,\mathrm{cm}$ and a width (i.e. Gaussian standard deviation) of $36.4\,\mathrm{cm}$. Our numerical results are shown in Figure 16, where the double bouncing is clearly visible. Moreover, the wave can be seen to propagate for a time after its encounter with the second turning point, even on the linear scale used for the plot.

# 5 Summary and conclusion

In this paper we have considered viscous dissipation of surface waves on fluids, with particular emphasis on its relevance to analogue gravity experiments. We began by considering rather general properties of dissipation, remarking that even in wave systems it shows much the same qualitative behavior as for a simple damped pendulum, including a bifurcation between underdamped and overdamped regimes. We noted that viscous dissipation manifests itself in two ways: as damping in the bulk due to friction between different layers of fluid, and as damping at the boundaries due to friction between the fluid and the floor or walls of its container. The former has a very simple analytic expression and is most relevant for short wavelengths; the latter, by contrast, has a rather complicated analytic expression and is most relevant for long wavelengths. Even so, the dissipation length associated with bulk viscosity at short wavelengths is typically much the smaller of the two, and the only one which can easily be less than the typical length scales of a flume. We thus focused on viscous dissipation in the bulk, whose simple analytic expression allows it to be easily incorporated into known wave equations.

We then turned our attention to particular experimental setups relevant to analogue gravity, in order to explore the relevance of viscous dissipation in that context. Unsurprisingly, it is most severe when short wavelengths are involved. Since the analogue Hawking effect can occur with purely gravity waves, it is less of an issue there, with the damping of waves during their propagation between the interaction region and the detector being relatively slow if the asymptotic flow velocity is large enough. However, the other experiments we considered involve very short wavelengths on the capillary branch, making them more susceptible to the effects of viscous dissipation: it makes capillary-regime black hole lasers unfeasible (at least in water), and it imposes quite severe restrictions on the design of analogue wormhole and double bouncing experiments.

Finally, we wish to note that, while we pointed out the existence of underdamped and overdamped regimes in our general discussion of dissipative effects, we then restricted ourselves to weak dissipation (i.e. the underdamped regime) when applying these effects to analogue gravity experiments. This ensures that dissipative effects do not fundamentally alter the non-dissipative physics. A gradual increase of the dissipation rate would of course reduce the visibility of any looked-for signal, and the optimized experimental parameters would evolve accordingly. However, if dissipation became too strong – indeed, if the overdamped regime were reached – we would enter a new physical regime where it is unlikely that we can speak of, say, Hawking radiation or even of wave propagation in any meaningful way. From the analogue gravity point of view, this strongly damped regime could be an interesting avenue for future research.

## Acknowledgements

S.R. was funded by ACI during a 4-month postdoctoral position at the University of Poitiers in early 2016, and thanks the University of Poitiers for their hospitality during that time. The work was funded by the French National Research Agency (ANR) through the grant HARALAB (ANR-15-CE30-0017-04). Support was also received from an FQXi grant of the Silicon Valley Community Foundation.

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
