# Peer review of "Viscous dissipation of surface waves and its relevance to analogue gravity experiments"

_SciPost Physics_

## Round 3 · Referee Report · Anonymous (Referee 1) · 2018-6-4

Strengths

1) The paper is relevant to design analogue gravity experiments. It elucidates the regimes where different effects may be observed and what are the optimal parameters to minimise the disrupting effects of viscous dissipation.

Weaknesses

1) the long and pedagogical review of the effects of dissipation in the harmonic oscillator, and surface waves in viscous fluids does not seem to be needed for what the authors actually do in the paper.

2) The equation studied in the manuscript lacks proper introduction

3) Setup of the simulations is not clear

Report

“Viscous dissipation of surface waves and its relevance to analogue gravity experiments” by S. Robertson and G. Rousseaux treats the design of analogue gravity experiments to minimise the effects of dissipation that disturb the desired outcome.

The authors first revisit the concept of dissipation in the widely known example of the harmonic oscillator in the underdamped, critical and overdamped regimes. They then use the example of one-dimensional acoustic waves in a gas to remind the reader that the stationary modes of dissipative waves behave as individual harmonic oscillators. The dispersion relation is written in two alternative forms that are well adapted for either forcing in time and dissipation in space, or for forcing in space and dissipation in time. The authors recover the underdamped and the overdamped regimes by taking the asymptotic regimes for large and small Reynolds number. They connect dissipation in space and time through the group velocity. They then describe another effect of dissipation, i.e. the introduction of dissipation causes instabilities in the vector potential in the direction of the axis of a rotating conducting cylinder.

The authors then revisit background on surface waves in fluids, first in the inviscid regime and then including viscous dissipation. In the inviscid regime, long wavelengths are dominated by gravity whereas short wavelengths are determined by capillarity. The dispersion relation holds for a uniformly moving fluid as well, with a Doppler-shift in the frequency between the lab and co-moving frames. They then review the effects of viscosity on surface waves. Viscosity causes energy dissipation through bulk as well as boundary effects, both reviewed by the authors. Additionally, viscosity causes corrections to the real part of the frequency, corresponding to alteration of the dispersion profile. This last effect introduces a minimum wavelength below which waves do not propagate (but this is not considered any further by the author’s own work). The authors demonstrate that when the fluid is in motion, waves of negative energy can appear. The authors then pass to discuss the Unruh model, which should model the high dissipation rate of capillary waves. If I understand correctly 3.5 is also a review of what is known about this model.

Section 4 contains the original results. The authors simulate numerically eq 22 and quantify dissipation in the regime that is of interest to reproduce the Hawking effect (amplification of incoming rays due to partial scattering into negative-energy outgoing waves) the black hole laser effect (trapping of self amplifying modes) and the wormhole traversal (whereby a specific regime allows rays to escape an apparent black hole or enter an apparent white hole). To study these analogue gravity experiments, the authors simulate numerically equation 22. The simulations indicate that in the regimes considered by the authors (1) dissipation is negligible for the Hawking effect; (2) even in optimal conditions, dissipation kills off the instability that causes the black hole laser effect in the capillary regime, whereas in the gravity regime viscous dissipation is unimportant; (3) distance between the black hole and the white hole needs to be shorter than the dissipative length in order to observe wormhole propagation - this makes it hard to observe it in water; (4) the authors find a regime where it is still possible to observe double bouncing, with parameters that can be realized experimentally (these optimal parameters are not far from those used in experiments).

My main criticism is that the long and pedagogical review of the effects of dissipation in the harmonic oscillator, and surface waves in viscous fluids does not seem to be needed for what the authors actually do in the paper. This is in contrast with the fact that equation (22), that the authors actually study, appears at page 14 quite abruptly. It is not immediately clear what is the goal of the simulations and what are the assumptions that go into their derivation.

The manuscript would greatly gain in clarity if it would start from paragraph 3.5, stating the regime that is relevant for the experiments, presenting the relevant equation, and mentioning the known results for special cases in an appendix. While the results are relevant for a proper design of the experimental realization of analogue gravity experiments, I cannot recommend the manuscript for publication in the present form.

Requested changes

Section 2 does not seem necessary, and section 3 can become a supplementary information, or shortened considerably to introduce the gravity vs capillary regimes of surface waves and the dissipation length. If section 3 remains in the main text, please state clearly the take home message for each paragraph.

Consider moving 3.5 as the first paragraph of section 4, as this is the equation studied throughout the manuscript.

The authors state that the weak bulk dissipation regime, neglecting friction caused by the boundary is the relevant regime to describe the experiments: is equation 22 an effective model that is valid in this regime or is it more general ? Is the goal of the simulations to account for a non-uniform velocity, and is this equation appropriate to this end ? What are the limits of this equation, could you comment on its relevance in practice ?

Is the example of dispersion relation in Figure 6 result of your simulations ? Is the previous discussion needed- given that the authors only focus on weak bulk dissipation ?

Minor comments: - Throughout the paper wave propagation in a static fluid is mentioned - do you mean stationary ? - please define phi before equation 4 - please use two different symbols for conductivity in equation (8) and frequency in eq (10) - they are currently both sigma. - page 14 take account of the high dissipation -> take into account the high dissipation - page 17: ‘in Figure 7 are plotted’ -> ‘Figure 7 shows’; ‘In the left panels are shown’->’the left panel shows’. There are several instances of this passive construct throughout the manuscript - page 24 “daming” -> damping - there are various typos that should come up in an automatic spell check

---

## Editorial Decision

awaiting_resubmission